# A Candidate Gene Cluster for the Bioactive Natural Product Gyrophoric Acid in Lichen-Forming Fungi

Garima Singh,[a,b,c] Anjuli Calchera,[a] Dominik Merges,[a] Henrique Valim,[a] Jürgen Otte,[a] Imke Schmitt,[a,b,d] Francesco Dal Grande[a,b,c]

[a]Senckenberg Biodiversity and Climate Research Centre (SBiK-F), Frankfurt, Germany
[b]LOEWE Center for Translational Biodiversity Genomics (TBG), Frankfurt, Germany
[c]Department of Biology, University of Padova, Padua, Italy
[d]Institute of Ecology, Diversity and Evolution, Goethe University, Frankfurt, Germany

**ABSTRACT** Natural products of lichen-forming fungi are structurally diverse and have a variety of medicinal properties. Despite this, they have limited implementation in industry mostly because the corresponding genes are unknown for most of their natural products. Here, we implement a long-read sequencing and bioinformatic approach to identify the putative biosynthetic gene cluster of the bioactive natural product gyrophoric acid (GA). Using 15 high-quality genomes representing nine GA-producing species of the lichen-forming fungal genus *Umbilicaria*, we identify the most likely GA cluster and investigate the cluster gene organization and composition across the nine species. Our results show that GA clusters are promiscuous within *Umbilicaria*, and only three genes are conserved across species, including the polyketide synthase (*PKS*) gene. In addition, our results suggest that the same cluster codes for different, but structurally similar compounds, namely, GA, umbilicaric-, and hiascic acid, bringing new evidence that lichen metabolite diversity is also generated through regulatory mechanisms at the molecular level. Ours is the first study to identify the most likely GA cluster and, thus, provides essential information to open new avenues for biotechnological approaches to producing and modifying GA and similar lichen-derived compounds. GA PKS is the first tridepside PKS to be identified.

**IMPORTANCE** The implementation of natural products in the pharmaceutical industry relies on the possibility of modifying the natural product (NP) pathway to optimize yields and pharmacological effects. Characterization of genes and pathways underlying natural product biosynthesis is a major bottleneck for exploiting the medicinal properties of the natural products. Genome mining is a promising and relatively cost- and time-effective approach to utilize unexplored NP resources for drug discovery. In this study, we identify the most likely gene cluster for the lichen-forming fungal depside gyrophoric acid in nine *Umbilicaria* species. This compound shows cytotoxic and antiproliferative properties against several cancer cell lines and is also a broad-spectrum antimicrobial agent. This information paves the way for generating GA analogs with modified properties by selective activation/deactivation of genes.

**KEYWORDS** biosynthetic genes, depsides, fungi, genome mining, long-read sequencing, microbial biotechnology, PKS phylogeny, secondary metabolites, *Umbilicaria*, genomics, lichen compounds, nonreducing PKSs, pharmaceutically relevant natural products

Natural products (NPs) and their derivatives/analogs constitute about 70% of modern medicines (1, 2). NPs alone, however, i.e. unmodified molecules as produced by organisms in nature, constitute only a small portion of this. The vast majority, about 60 to 65%, are derivatives and analogs of naturally occurring substances, synthesized through biotechnological or synthetic approaches (2, 3). The exploitation of NPs in the pharmaceutical industry relies on the possibility of modifying NP pathways in order to optimize

Address correspondence to Garima Singh, garima.singh@unipd.it, or gsingh458@gmail.com.

The authors declare no conflict of interest.

yields and to achieve the desired pharmacological effects. Culture-dependent approaches to identify/produce NPs are labor-intensive and time-consuming and are not successful for every organism (4, 5). As a result, the biosynthetic potential of many biosynthetically prolific organisms remains untapped. Information on the genetic background and mechanisms of NP synthesis may thus contribute to fast-tracking NP-based drug discovery (2, 6).

Lichens, symbiotic organisms composed of fungal and photosynthetic partners (green algae or cyanobacteria or both at the same time) (7–9), are a treasure chest of NPs (10–12). Lichen compounds have great pharmacological potential, encompassing antimicrobial, antiproliferative, cytotoxic, and antioxidant properties (11, 13–16). So far, about 1,000 NPs with great structural and functional diversity have been reported from lichen-forming fungi (LFF), and about 300 to 400 have been screened for bioactivity (11). However, the genetic background of more than 97% of lichen NPs is unknown (17–20). This is because, there are various major bottlenecks in using lichen NPs in the pharmaceutical industry, including low yield in nature, slow growth, and/or tedious isolation/culturing methods. Targeted genome mining approaches integrate the latest DNA sequencing technologies with computational advancements and large, publicly available databases of already characterized biosynthetic gene clusters (BGCs) to identify genes coding for NPs (1, 21, 22). This approach combines genome mining with the expected genetics of the NP to narrow down the candidate biosynthetic genes.

*In silico* approaches for linking natural products with their respective BGCs—genomic clusters of biosynthetic-related genes typically found in fungi (23–25)—are becoming more common in LFF due to the increased availability of genomic resources and databases (10, 17, 20), improvement of detection software and genome mining tools, stabilizing polyketide synthase (PKS) phylogenies, and information gained from recent successes in the heterologous expression of *PKS* from LFF (17, 18). For instance, the clade "group I, PKS16" from Kim et al. (17) is associated with the biosynthesis of orsellinic acid derivatives (orcinol depsides and depsidones), such as lecanoric acid (18), grayanic acid (19), physodic acid, and olivetoric acid (20), whereas the clade "group IX, PKS23" from Kim et al. (17) is associated with the biosynthesis of methylated orsellinic acid derivatives (*β*-orcinol depsides and depsidones) such as atranorin. The cluster linked to usnic acid biosynthesis is also fairly well studied (26, 27) and corresponds to "group VI, PKS8" from Kim et al. (17).

Here, we combine high-throughput long-read sequencing with a comparative genomics approach to identify the putative cluster(s) linked to the synthesis of gyrophoric acid (GA). GA is a natural product synthesized by several LFF species. It has a broad spectrum of bioactivity, such as anticancer and antimicrobial, and industrially-relevant properties, including usage as dyes (15, 28–30). However, the molecular mechanism and genetics of its synthesis remain unknown. Identification of the GA cluster would facilitate its production via biotechnology to optimize yield as well as to generate GA analogs with the desired pharmaceutical effect. For this study, we chose nine species of GA producers belonging to the lichen-forming fungal genus *Umbilicaria* (Table 1). GA is the most characteristic compound of this genus and is found at high concentrations in all of the chosen species (28, 31–33). It is a depside containing three orsellinic acid rings joined together by ester bonds (Fig. 1). Apart from GA, several other structurally related depsides, such as umbilicaric acid, lecanoric acid, and hiascic acid (Fig. 1), have also been reported from *Umbilicaria* spp. (31, 32, 34), but these usually constitute a minor fraction (<10%) of the total NPs detected via high-pressure liquid chromatography (HPLC) (Fig. 1).

In the present study, we assembled highly contiguous long-read-based genomes of the nine species of the genus *Umbilicaria*, identified their biosynthetic gene clusters, and singled out the candidate genes linked to GA biosynthesis.

## RESULTS

**Genome sequencing, assembly, and annotation.** The genome quality stats and assembly reports of all of the genomes used for this study are presented in Table 1.

**Total BGCs and phylogenetic analysis.** A total of 406 BGCs and 236 PKSs were identified in 15 *Umbilicaria* genomes, representing nine species (Table 2). Out of 236, 122 were nonreducing PKSs (NR-PKSs), 86 were reducing PKSs, 16 were type-III PKSs, and 12 were partial or

**TABLE 1** Genome quality and annotation statistics

| Taxon | Sample ID | Ccs HiFi yield (%) | No. of Scaffolds | $N_{50}$ | Completeness (%) | Assembly size (Mb) | No. of genes | No. of proteins | Genome accession |
|---|---|---|---|---|---|---|---|---|---|
| *Umbilicaria deusta* 1 | TBG_2334 | 47.86 | 44 | 1.67 | 97.6 | 40.9 | 8,949 | 8,857 | JALILR000000000 |
| *U. deusta* 2 | TBG_2335 | 43.54 | 42 | 1.87 | 90.2 | 37.4 | 8,194 | 8,049 | NA |
| *U. freyi* 1 | TBG_2329 | 47.39 | 107 | 2.58 | 95.7 | 47.5 | 10,156 | 10,065 | JALILQ000000000 |
| *U. freyi* 2 | TBG_2330 | 46.41 | 54 | 2.04 | 85.9 | 50 | 8,848 | 8,773 | NA |
| *U. grisea* | TBG_2336 | 42.54 | 40 | 1.83 | 96.9 | 44.43 | NA | NA | JALILX000000000 |
| *U. hispanica* 1 | TBG_2322 | 38.71 | 130 | 3.13 | 96.8 | 43.4 | 9,111 | 9,021 | NA |
| *U. hispanica* 2 | TBG_2337 | 54.22 | 53 | 4.23 | 97.3 | 48.6 | 8,781 | 8,696 | JALILS000000000 |
| *U. phaea* 1 | TBG_1111 | NA[a] | 47 | 1.54 | 96.5 | 35.1 | 7,659 | 7,576 | NA |
| *U. phaea* 2 | TBG_1112 | NA | 38 | 1.22 | 96.5 | 35.55 | 7,681 | 7,628 | JALILT000000000 |
| *U. pustulata* 1 | TBG_2333 | 33 | 26 | 2.62 | 97.3 | 37.7 | 9,569 | 9,503 | NA |
| *U. pustulata* 2 | TBG_2345 | 32.26 | 31 | 2.36 | 96.8 | 35.7 | 8,790 | 8,740 | JALILU000000000 |
| *U. spodochroa* 1 | TBG_2434 | 34.20 | 130 | 9.93 | 97.0 | 44.3 | 8,791 | 8,705 | JALILV000000000 |
| *U. spodochroa* 2 | TBG_2435 | 40.93 | 97 | 1.25 | 97.1 | 40.1 | 8,612 | 8,507 | NA |
| *U. subpolyphylla* 1 | TBG_2323 | 41.14 | 190 | 1.55 | 99.6 | 58.2 | 16,993 | 16,852 | NA |
| *U. subpolyphylla* 2 | TBG_2324 | 33.68 | 39 | 1.52 | 97.6 | 31.8 | 8,556 | 8,410 | JALILW000000000 |
| *Dermatocarpon miniatum* 1 | TBG_2326 | 29.36 | 26 | 5.08 | 98.1 | 63.5 | 9,273 | 9,189 | JALILY000000000 |
| *D. miniatum* 2 | TBG_2331 | 26.28 | 22 | 4.25 | 98.4 | 49.8 | 7,938 | 7,871 | NA |

[a]NA, not applicable.

unclassified because the core *PKS* was fragmented and the characteristic domains were missing (Table 2).

Four NR-PKSs were common to all species, namely, PKS15, PKS16, PKS20, and a novel PKS (forming a monophyletic, supported clade with PKS33) (Fig. 2). Only one NR-PKS per species formed a supported monophyletic clade with PKS16 (group I, i.e., orsellinic acid, depside, and depsidone NR-PKSs) (Fig. 2). No PKS from *Dermatocarpon miniatum* grouped within the PKS16 clade, which is expected, as *D. miniatum* does not produce orsellinic acid-based compounds.

**Gyrophoric acid cluster.** The cluster most likely associated with GA synthesis is the cluster containing *PKS16* (Fig. 2), as (i) it is present in all *Umbilicaria* spp., (ii) it contains an *NR-PKS*, and (iii) it forms a monophyletic group with the clade "group I, PKS 16" from Kim et al. (17).

Out of 15 genes (in the GA cluster of *Umbilicaria deusta*) (Fig. 3A), antiSMASH identified two genes, *PKS* and *cyt P450*, whereas other genes were identified as coding for proteins of unknown function by antiSMASH as well as by InterProScan. NCBI conserved domain search (CDS) identified the additional 10 genes of the candidate GA cluster and estimated their putative function based on their domain motifs (Fig. 3B). Specifically, these genes code for proteins involved in transcription regulation, oxidation, hydrolysis, and protein-protein interaction/trafficking. Conserved domains were not detected in three genes (genes 5, 11, and 13; Fig. 3B), and two genes (genes 7 and 14; Fig. 3B) had conserved domains belonging to the DUF (domain of unknown function) superfamily.

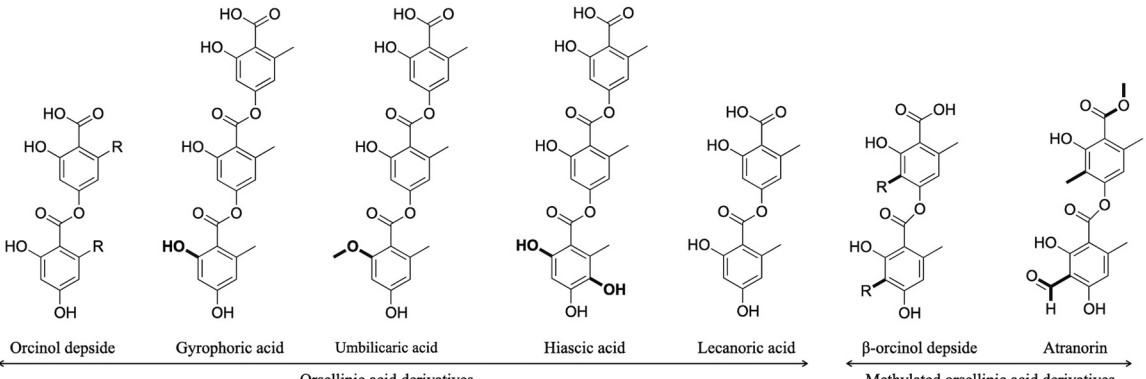

| Orcinol depside | Gyrophoric acid | Umbilicaric acid | Hiascic acid | Lecanoric acid | β-orcinol depside | Atranorin |

Orsellinic acid derivatives                                    Methylated orsellinic acid derivatives

**FIG 1** Chemical structures and nomenclature. Structure of a lichen depside, atranorin, GA, and other depsides produced by *Umbilicaria* spp.

**TABLE 2** Biosynthetic gene clusters and PKSs identified in *Umbilicaria* spp[a]

| Taxon | Sample ID | No. BGCs | Total no. PKSs | No. NR-PKSs | No. red-PKSs | No. unclassified/ fragmented | No. T3PKS |
|---|---|---|---|---|---|---|---|
| *Umbilicaria deusta* 1 | TBG_2334 | 33 | 24 | 11 | 10 | 1 | 2 |
| *U. deusta* 2 | TBG_2335 | 30 | 21 | 11 | 9 | 1 | 0 |
| *U. freyi* 1 | TBG_2329 | 25 | 9 | 5 | 3 | 0 | 1 |
| *U. freyi* 2 | TBG_2330 | 23 | 10 | 4 | 5 | 0 | 1 |
| *U. grisea* | TBG_2336 | 20 | 13 | 7 | 4 | 1 | 1 |
| *U. hispanica* 1 | TBG_2322 | 24 | 13 | 9 | 3 | 0 | 1 |
| *U. hispanica* 2 | TBG_2337 | 25 | 14 | 9 | 4 | 0 | 1 |
| *U. phaea* 1 | TBG_1111 | 22 | 14 | 6 | 7 | 0 | 1 |
| *U. phaea* 2 | TBG_1112 | 19 | 11 | 6 | 4 | 0 | 1 |
| *U. pustulata* 1 | TBG_2333 | 31 | 18 | 8 | 9 | 0 | 1 |
| *U. pustulata* 2 | TBG_2345 | 27 | 17 | 9 | 6 | 1 | 1 |
| *U. spodochroa* 1 | TBG_2434 | 27 | 14 | 8 | 4 | 1 | 1 |
| *U. spodochroa* 2 | TBG_2435 | 27 | 16 | 9 | 5 | 1 | 1 |
| *U. subpolyphylla* 1 | TBG_2323 | 30 | 14 | 7 | 4 | 1 | 2 |
| *U. subpolyphylla* 2 | TBG_2324 | 20 | 10 | 5 | 4 | 0 | 1 |
| *U. muhlenbergii* | NA | 23 | 18 | 8 | 5 | 5 | 0 |
| *Dermatocarpon miniatum* 1 | TBG_2326 | 29 | 13 | 3 | 8 | 2 | 0 |
| *D. miniatum* 2 | TBG_2331 | 32 | 11 | 4 | 6 | 1 | 0 |

[a]Red-PKSs, reducing PKSs; NR-PKSs, nonreducing PKSs; T3PKSs, type III PKSs.

We inferred the synteny of the *U. deusta* GA cluster with the GA clusters of all other *Umbilicaria* spp. to estimate homology between them (Fig. 4). In addition, we also examined the synteny of the *U. deusta* GA cluster with the other clusters involved in the synthesis of orsellinic-acid-derivative compounds—olivetoric acid, grayanic acid, and orsellinic acid cluster (Fig. 4). The synteny plots show that GA clusters are highly homologous among *Umbilicaria deusta*, *U. freyi*, *U. grisea*, *U. phaea*, and *U. subpolyphylla*, whereas between GA and grayanic acid cluster and GA and olivetoric acid cluster only the *PKS* genes are homologous (Fig. 4|). The other genes of the clusters involved in the production of orsellinic acid derivatives are not conserved among the genera examined, i.e., *Umbilicaria* spp., *Cladonia grayi*, and *Pseudevernia furfuracea*. The orsellinic acid cluster from *Aspergillus nidulans* showed almost no homology to the GA cluster.

**BGC clustering: BiG-SCAPE and CORASON.** BGC sequence similarity networks group gene clusters at multiple hierarchical levels. This analysis implements a "glocal" alignment mode that groups both complete and fragmented BGCs. The BGCs forming a supported monophyletic clade to PKS16 (group I) were then analyzed for conservation across species using CORASON. The CORASON analysis also showed that only the following three genes in the cluster were shared among the studied *Umbilicaria* species: the core *PKS* and the two genes present upstream and downstream of the core gene (Fig. 5).

We used CORASON plot in order to display the GA clusters of *Umbilicaria* spp. (Fig. 5). There are 9 to 15 genes present in the putative GA cluster of the species studied. The *PKS* and genes upstream and downstream of it are present in all *Umbilicaria* spp. As the number of genes present in the GA cluster varies among the *Umbilicaria* spp., we consider the *U. deusta* GA cluster as a representative of *Umbilicaria* GA clusters for this study (Fig. 3A). The gene numbers refer to the genes in the GA cluster of *U. deusta* (Fig. 3A).

## DISCUSSION

**Gyrophoric acid PKS.** We identified only one PKS as the most likely GA PKS (Fig. 2).

So far, the BGCs associated with the biosynthesis of the following lichen depsides and depsidones have been experimentally identified: atranorin (17), lecanoric (18), and grayanic acid (19). These studies demonstrate that the PKS alone is capable of synthesizing the backbone depside, whereas modifications, such as methylation and oxidation, are made by enzymes coded by other genes of the cluster after the release of the depside from the PKS. For instance, the synthesis of atranorin involves at least three genes, but the depside backbone is coded by the *PKS* (17). The other two genes, an *O*-methyltransferase (OMT) and a *cyt P450*, methylate the carboxyl group and oxidize the methyl group (into

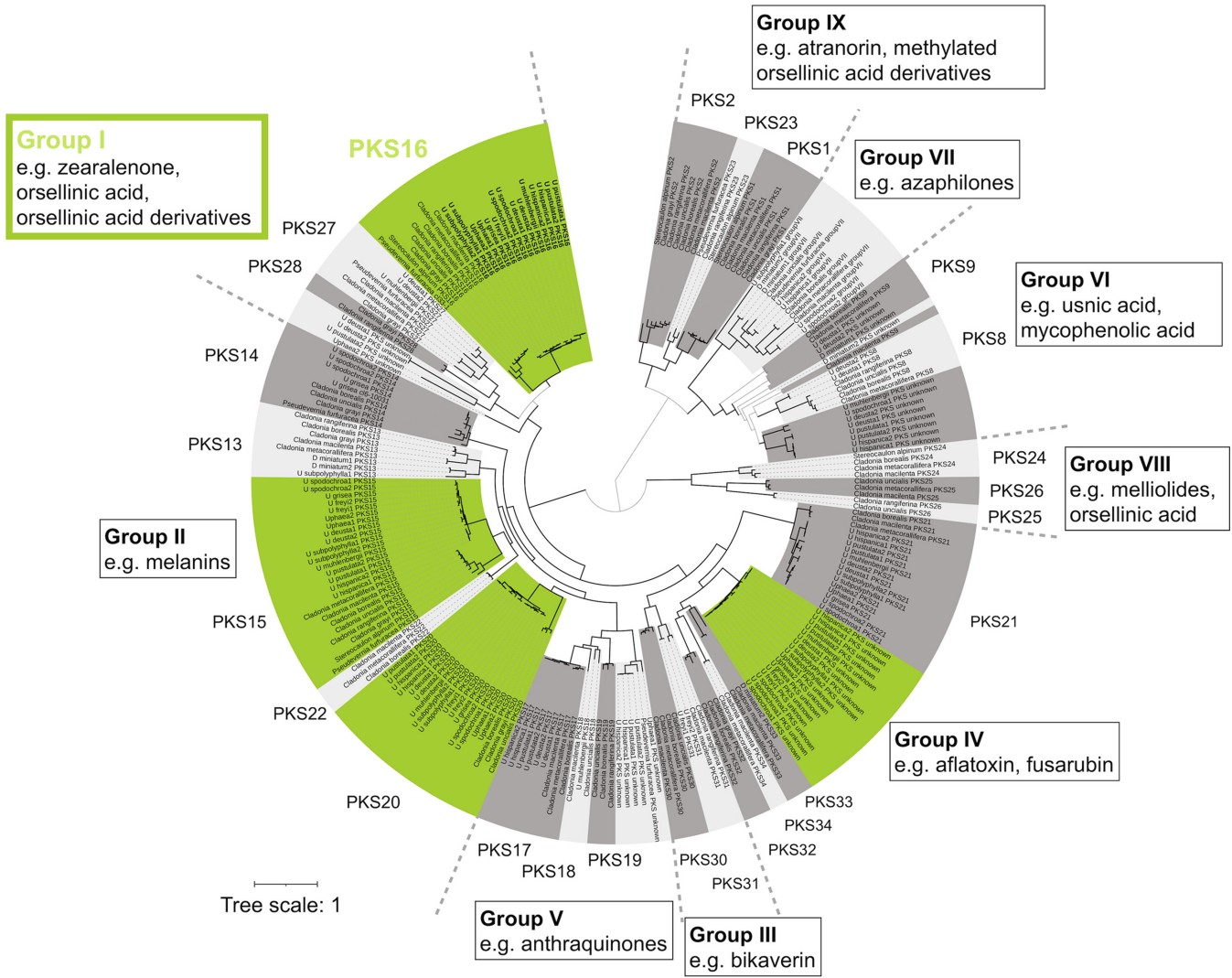

**FIG 2** NR-PKS phylogeny of lichen-forming fungi. This is a maximum-likelihood tree based on amino acid sequences of NR-PKSs from nine *Umbilicaria* spp., six *Cladonia* spp., *Dermatocarpon miniatum*, *Stereocaulon alpinum*, and *Pseudevernia furfuracea*. Branches in bold indicate bootstrap support >70%. Green color clades represent the PKSs common to all nine *Umbilicaria* spp. used in this study. PKS groups are based on Kim et al. (17).

-CHO), respectively, to produce the final product atranorin (Fig. 1). As GA does not have side-chain modifications (Fig. 1), we propose that the PKS alone is capable of producing the final product, i.e. GA.

The depside *PKSs* identified so far code for didepsides, i.e. compounds that contain two phenolic rings joined with an ester bond, e.g., atranorin, lecanoric-, and olivetoric acid (17, 18, 20). Ours is the first study to identify the most likely *PKS* associated with a tridepside synthesis, i.e., three phenolic rings joined with two ester bonds. Our study suggests that the *PKS* genes coding for a didepside and a tridepside are highly homologous (Fig. 4).

**GA cluster in *Umbilicaria* spp.** The most likely GA cluster contains about 9 to 15 genes in different *Umbilicaria* spp. (Fig. 4 and 5). Interestingly, only the following three genes are conserved across the species analyzed: the *PKS*, a gene having a conserved domain of unknown function present upstream, and a hydrolase present downstream of the *PKS* (Fig. 4 and 5). This suggests that these three genes form an integral part of the GA cluster, whereas the other genes are facultative among GA producing *Umbilicaria* spp. The facultative genes in the GA cluster code for enzymes involved in transcription regulation, oxidation, hydrolysis, and protein-protein interaction/trafficking, indicating that they play a role in species-specific transcription regulation, transport, or modification (Fig. 3). Differences in gene content among the clusters synthesizing the same compound have been reported before. For instance, the usnic

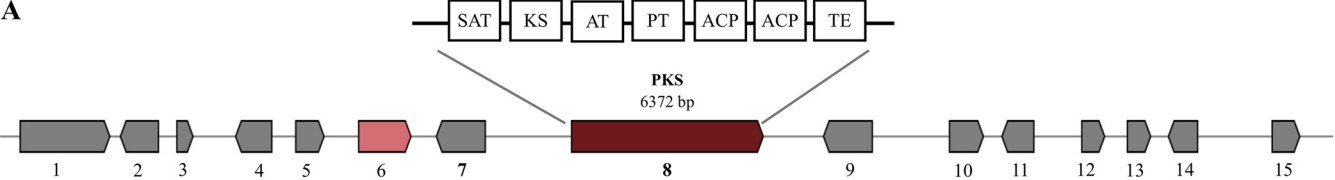

**FIG 3** (A) Gyrophoric acid cluster from *Umbilicaria deusta* as predicted by antiSMASH. Colored boxes indicate genes. Genes in gray represent genes coding for unknown proteins as predicted by the antiSMASH. (B) Putative functions of the genes of the gyrophoric acid cluster based on InterProScan and NCBI CDS search. The numbers correspond to the gene numbers of *U. deusta* in panel A.

| Gene | Domain (NCBI CDS search) | Putative function |
|---|---|---|
| 1 | Hook domain-containing proteins superfamily | adaptor protein and protein trafficking |
| 2 | Oxidase | oxidase |
| 3 | Ubl_ATG8 | transcriptional regulation, cell cycle control and mediating protein-protein interaction. |
| 4 | Mito_carr | mitochondrial carrier protein |
| 5 | No CDS detected | unknown |
| 6 | CYP56-like, cytochrome P450 family 56-like fungal cytochrome P450s | oxidation |
| 7 | DUF3533 | protein of unknown function DUF45; This protein has no known function. |
| 8 | PKS | polyketide synthesis |
| 9 | metallo-dependent_hydrolases super family | hydrolase |
| 10 | Acyl CoA binding protein | acyl CoA binding protein (ACBP) binds thiol esters of long fatty acids and coenzyme A in a one-to-one binding mode with high specificity and affinity. |
| 11 | No CDS detected | unknown |
| 12 | BTB_POZ super family | protein-protein interaction: diverse functions, such as transcriptional regulation, chromatin remodeling, protein degradation and cytoskeletal regulation. |
| 13 | No CDS detected | unknown |
| 14 | DUF45 super family | protein of unknown function DUF45; This protein has no known function. |
| 15 | LicD super family | nucleotidyltransferase superfamily: involved in phosphorylcholine metabolism |

acid cluster differs in its gene content among taxa, and out of 8 to 15 genes belonging to the cluster, only *PKS* and *cyt P450* are conserved (26). These differences could be responsible for species-specific BGC regulation or modifications to the protein released by the PKS.

The most likely GA cluster contains a *cyt P450* (Fig. 3A), which has been associated with oxidative function (17, 19, 20, 27). However, the location and orientation of the *cyt P450* in the *Umbilicaria* GA clusters are different from those of a cluster which requires an active *cyt P450* for the production of the final compound, i.e., grayanic acid, usnic acid, and atranorin cluster (Fig. 4) (17, 19, 26). In the GA cluster the *cyt P450* is not located next to the *PKS* and has the same orientation as the *PKS*, whereas in the grayanic acid, usnic acid, and atranorin cluster, *cyt P450* lies next to the *PKS* in the opposite orientation (Fig. 4). For instance, grayanic acid synthesis (in *Cladonia grayi*) involves the synthesis of the depside intermediate by PKS followed by oxidation of the released depside into depsidone by cyt P450 (19). Such organization is suggestive of genes being regulated and coexpressed by the same promoter (17, 19). *PKS* and *cyt P450* form an integral part of depsidone synthesis (35), whereas the depside is coded by the *PKS* alone, with the exception of the side-chain modifications (17, 18). Therefore, despite being part of the GA cluster, *cyt P450* does not seem to be involved in GA synthesis or in the synthesis of umbilicaric and/or lecanoric acid reported from *Umbilicaria* spp. analyzed in this study. The synthesis of hiascic acid, however, would require the hydroxylation of a methyl group by a cyt P450 enzyme after the depside is released from the PKS (Fig. 1, the OH group in bold in hiascic acid). The lower amounts of hiascic acid found in *Umbilicaria* thalli, compared to GA, could be due to the fact that the *cyt P450* may not be coexpressed with the *PKS*.

Although the GA cluster shows varied degrees of homology among *Umbilicaria* spp., with certain genes present only in some species, the gyrophoric acid PKS is highly homologous among *Umbilicaria* (Fig. 4). This is expected, as the PKSs are involved in the synthesis of the same compound. Interestingly, the GA PKS is also homologous to the olivetoric acid PKS and grayanic acid PKS, suggesting that the depside/depsidone *PKS* are homologous even among distantly related taxa in lichen-forming fungi. The GA was not homologous to the orsellinic acid *PKS* from *Aspergillus nidulans* (Fig. 4), suggesting different evolutionary trajectories despite being involved in the synthesis of orcinol-derived compounds.

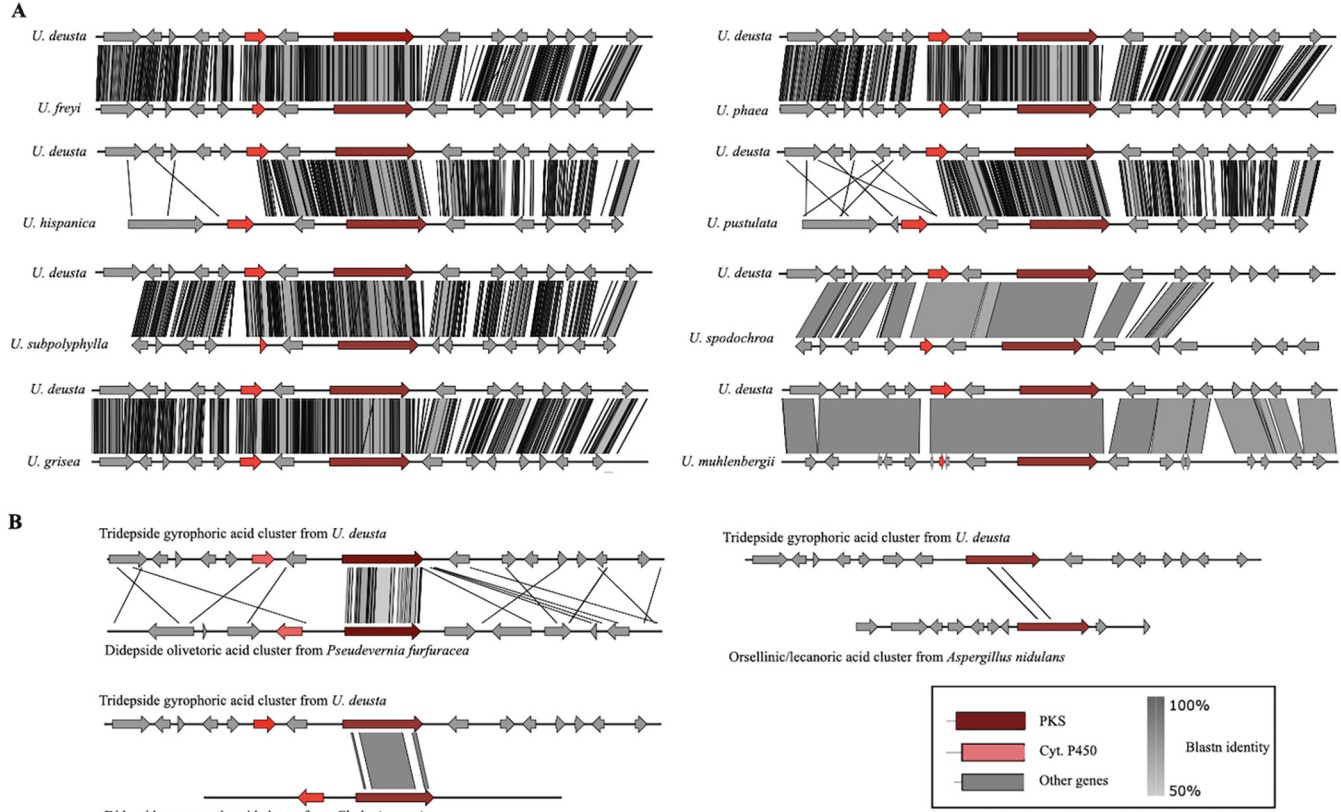

**FIG 4** Synteny plots based on tBLASTn showing the homology and synteny between the putative gyrophoric acid clusters *U. deusta* and other *Umbilicaria* spp. (A) and between the gyrophoric acid cluster from *U. deusta* and the grayanic acid cluster from *Cladonia grayi*, the olivetoric acid cluster from *Pseudevernia furfuracea*, and the orsellinic acid cluster from *Aspergillus nidulans* (B). All of the PKSs are highly homologous to the GA PKS and have the same domains as the GA PKS: SAT-KS-AT-PT-ACP-ACP-TE.

Our study provides novel insights into GA cluster composition and organization across different species (Fig. 5). This information is crucial in order to open the way for future genetic manipulation of the GA biosynthetic pathway that may be aimed at increasing structural diversity and/or yield of the products, as well as at generating analogs with novel properties.

**One cluster, different compounds.** Variation in cluster composition reflects the potential to produce diverse NPs. Apart from GA, other depsides related in structure to GA, i.e., lecanoric acid, umbilicaric acid, and hiascic acid (Fig. 1), are often reported from *Umbilicaria* spp. as minor metabolites (31). Interestingly, we found only one orcinol depside *PKS* in *Umbilicaria* spp. (Fig. 2). This strongly indicates that all of the *Umbilicaria* depsides are coded by the same PKS cluster. One cluster coding for different, structurally-related compounds has also been reported previously (20, 36, 37). For instance, in the case of the antifungal drug caspofungin acetate, a semisynthetic derivative of the NP pneumocandins from the fungus *Glarea lozoyensis*, selective inactivation of different genes in this biosynthetic gene cluster generates 13 different analogues, some of them with elevated antifungal activity relative to the original compound and its semisynthetic derivative (38). Similarly, the aspyridone biosynthetic cluster from *Aspergillus nidulans* produces eight different compounds in a heterologous host (37). These studies show that a single PKS cluster is capable of producing different compounds depending upon which genes are coexpressed and on the available starters. In lichens, a single *PKS* has been associated with the synthesis of olivetoric and physodic acid (20), and the same *PKS* has been shown to be involved in the synthesis of lecanoric acid in a heterologous host (18). We propose that the same PKS cluster is most likely involved in the synthesis of GA, umbilicaric acid (an additional methyl group) (Fig. 1), hiascic acid (additional hydroxyl group) (Fig. 1),

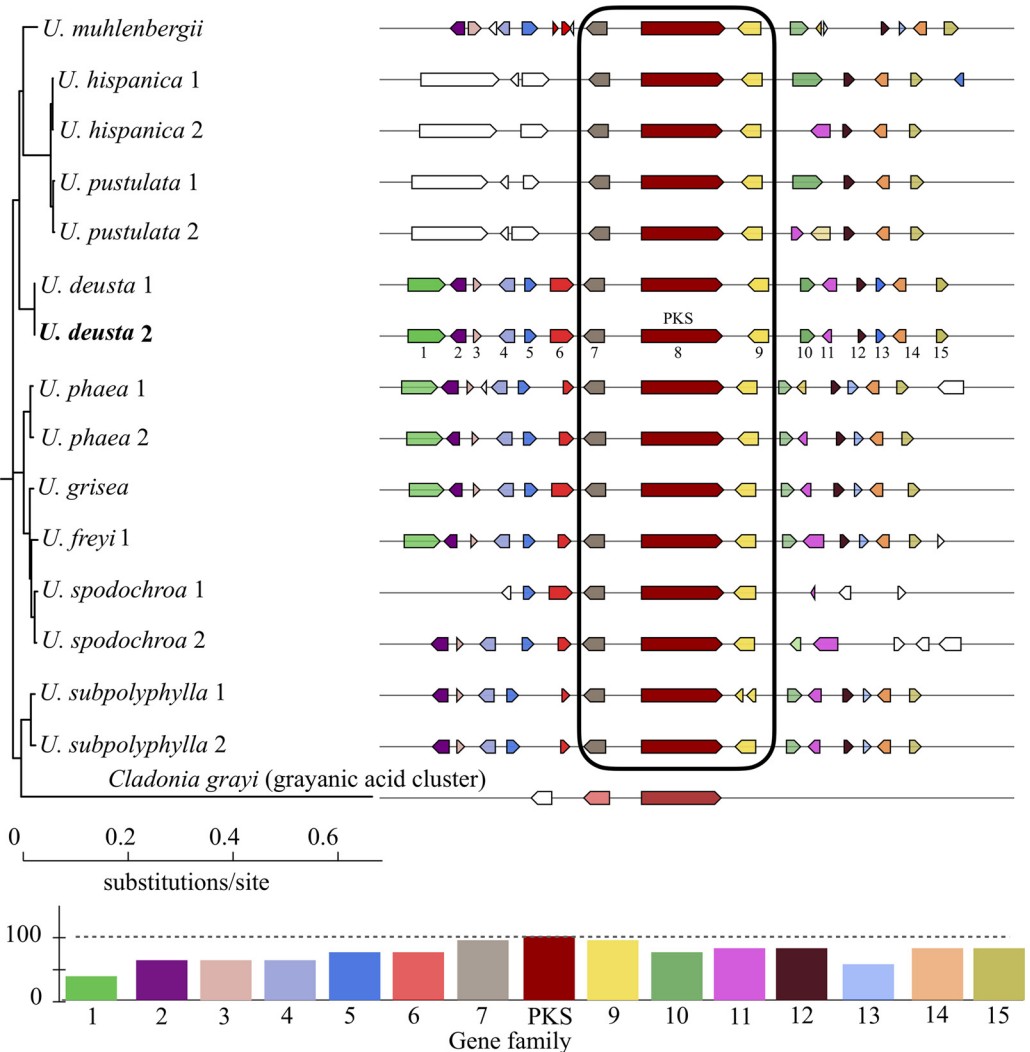

**FIG 5** CORASON-based PKS phylogeny to elucidate evolutionary relationships and cluster organization of the GA cluster in *Umbilicaria* spp. The bar plot below depicts the percentage of *Umbilicaria* species in which a particular gene is present. The color of the genes is white when the homology to the genes of the other gyrophoric acid clusters is low.

and lecanoric acid (didepside with no side chains) (Fig. 1) in *Umbilicaria*. It is possible, however, that in nature only GA is synthesized in members of the genus *Umbilicaria*, and the cooccurring minor compound lecanoric acid is a hydrolysis product of GA (39).

Interestingly, although umbilicaric acid is reported from some *Umbilicaria* species (*U. grisea*, *U. freyi*, *U. muhlenbergii*, and *U. subpolyphylla* [31, 40]), *O*-methyltransferase (OMT) was not identified in the depside-related BGC of any *Umbilicaria* species (Fig. 3A and B). OMT would be required for the methylation of oxygen to produce umbilicaric acid (Fig. 1). Its absence from depside-related BGCs suggests that an external OMT, e.g., from other BGCs, might be involved in the production of umbilicaric acid in *Umbilicaria*. This could explain the lower amounts of umbilicaric acid compared to GA found in these species (31). In contrast, when the *O*-methylated compound is the major secondary metabolite, as in the case of grayanic acid and atranorin, OMT is an integral part of the BGC and is coexpressed along with the other crucial genes for grayanic acid production, i.e., *PKS* and *cyt P450* (17, 19).

**Future perspectives.** Advances in long-read sequencing and in computational approaches to genome mining not only enable linking biosynthetic genes to NPs but also provide an overview of the entire gene cluster composition and organization.

Ours is the first study to identify the most-likely GA cluster, which is essential for opening up avenues for biotechnological approaches to producing and modifying this compound and possibly other lichen compounds. In particular, this information can be applied to generate novel NP analogs with improved pharmacological properties via synthetic biology, biotechnology, and combinatorial biosynthesis approaches. This paves the way to an entirely new horizon in terms of utilizing these understudied taxa for pharmacological industry and drug discovery.

## MATERIALS AND METHODS

**Sampling and data set.** We collected samples of the following eight *Umbilicaria* species for genome sequencing: *U. deusta*, *U. freyi*, *U. grisea*, *U. subpolyphylla*, *U. hispanica*, *U. phaea*, *U. pustulata*, and *U. spodochroa* (see voucher information in Table S1 in the supplemental material). When possible, we sequenced two samples of the same species collected in different climatic zones. This was done to consider the possible intraspecific variation in BGC content as recently shown in Singh et al. (34). The genome of *U. muhlenbergii* was downloaded from the JGI database. In addition, we sampled *Dermatocarpon miniatum* as a control, as it does not produce depsides/depsidones.

**DNA extraction, library preparation, and genome sequencing.** Lichen thalli were thoroughly washed with sterile water and checked under the stereomicroscope for the presence of possible contamination and other lichen thalli. DNA was extracted from all of the samples using a cetyltrimethylammonium bromide (CTAB)-based method (41) as presented in reference 42.

SMRTbell libraries were constructed according to the manufacturer's instructions of the SMRTbell Express prep kit v. 2.0 following the low DNA input protocol (Pacific Biosciences, Menlo Park, CA). Total input for samples was approximately 170 to 800 ng. Ligation with T-overhang SMRTbell adapters was performed at 20°C for 1 h or overnight. Following ligation, the libraries were purified with a 0.45× or 0.8× AMPure PB bead cleanup step. The subsequent size selection step to remove SMRTbell templates of <3 kb was performed with 2.2× of a 40% (vol/vol) AMPure PB bead working solution.

SMRT sequencing was performed on the Sequel system II with the Sequel II sequencing kit 2.0 using the continuous long read (CLR) mode or the circular consensus sequencing (CCS) mode, 30 h movie time with no preextension and Software SMRTLINK 8.0. Each metagenomic library was sequenced on one SMRT cell at the Medical Center Nijmegen (the Netherlands) or at MPI Dresden.

**Genome assembly and annotation.** The continuous long reads (i.e., CLR reads) from the PacBio Sequel II CLR run were first processed into highly accurate consensus sequences (i.e., HiFi reads) using PacBio tool CCS v5.0.0 with default parameters (https://ccs.how). HiFi reads were then assembled into contigs using the assembler metaFlye v2.7 (43). The resulting contigs were scaffolded with LRScaf v1.1.12 (https://github.com/shingocat/lrscaf; 44). The scaffolds were then taxonomically binned to extract Ascomycota reads with blastx using DIAMOND (–more-sensitive –frameshift 15 –range-culling) on a custom database and following the MEGAN6 Community Edition pipeline (45). All scaffolds assigned to Ascomycota were extracted as to represent the *Umbilicaria* spp. Assembly statistics, such as number of contigs, total length, and $N_{50}$ were accessed with Assemblathon v2 (46) (Table 1). The completeness of the mycobiont bins (i.e., the fungal genomes) was estimated using benchmarking universal single-copy orthologs (BUSCO) analysis in BUSCO v4 (47).

**Identification of biosynthetic gene clusters.** Functional annotation of genomes, including genes, proteins, and BGC prediction (antiSMASH; antibiotics & SM Analysis Shell, v5.0) was performed with scripts implemented in the funannotate pipeline (48, 49). First, the genomes were masked for repetitive elements, and then the gene prediction was performed using BUSCO2 to train Augustus and self-training GeneMark-ES (47, 50). Functional annotation was done with InterProScan (51), egg-NOG-mapper (52, 53), and BUSCO (47) with ascomycota_db models. Secreted proteins were predicted using SignalP (54) as implemented in the funannotate "annotate" command.

To complement the functional annotation of genes by InterProScan, egg-NOG-mapper, and SignalP, we performed NCBI CDS searches on the genes of the putative gyrophoric acid cluster using the nucleotide sequences of the genes.

**Phylogenetic analyses.** To search for PKSs involved in the synthesis of GA, we extracted the amino acid sequences of all of the NR-PKS from the BGCs predicted by the antiSMASH for the *Umbilicaria* spp. and *Dermatocarpon miniatum* (see Table S2 in the supplemental material). The sequences were aligned using MAFFT as implemented in Geneious v5.4 (55, 56). Gaps were treated as missing data. The maximum likelihood search was performed on the aligned sequences with RAxMLHPC BlackBox v8.1.11 (57) on the Cipres Scientific gateway (58). We then performed a phylogenetic analysis by incorporating these amino acid sequences into the most comprehensive PKS data set currently available (Table S2) (17, 20). This data set comprises NR-PKS sequences of the following species downloaded from previous publications and public databases: *Cladonia borealis*, *C. grayi*, *Cladonia macilenta*, *Cladonia metacorallifera*, *Cladonia rangiferina*, *Cladonia uncialis*, *Pseudevernia furfuracea*, *Stereocaulon alpinum*, and *Umbilicaria muhlenbergii*. The final data set contains amino acid sequences of 229 NR-PKSs from 18 species belonging to five LFF genera. Phylogenetic trees were visualized using iTOL (59).

**Annotation of PKSs.** *Umbilicaria* PKSs were named according to the clustering with preannotated PKSs in the phylogeny. NR-PKSs have been categorized into nine groups based on phylogenetic clustering and broad category of the protein coded by them (17). For instance, group I comprises PKSs involved in the synthesis of zearalenone, orsellinic acid, and its derivative compounds, and group II consists of PKSs coding for melanins.

Each of the nine groups contains several PKSs based on the supported phylogenetic clades and

protein sequence similarity. For this study, we included the following 25 NR-PKSs: PKS1, PKS2, PKS8, PKS9, PKS13 to PKS28 (total 16), and PKS30 to PKS34 (total 5). Each PKS represents a supported monophyletic clade within a group in the NR-PKS phylogeny (17). To summarize, *Umbilicaria* PKSs were annotated and named according to phylogenetic clustering with the preannotated NR-PKS sequences of *Cladonia* spp., *Pseudevernia furfuracea,* and *Stereocaulon alpinum*, downloaded from previous publications and public databases (17, 19, 20). PKS16 and PKS23 have been suggested to be involved in the synthesis of depsides, the chemical category of GA. The PKSs responsible for the synthesis of *β*-orcinol depsides such as atranorin are PKS23, whereas those involved in the synthesis of orcinol-depsides, such as grayanic acid and olivetoric acid are PKS16 (19, 20). GA is an orcinol depside; therefore, the corresponding PKS(s) would be PKS16.

**Selecting candidate gene clusters linked to GA biosynthesis.** We used the following criteria to select the candidate gene cluster associated with GA synthesis in *Umbilicaria*: (i) it must contain an NR-PKS (as some of the structural features of an NP can be directly inferred from the domain architecture of the *PKS*: *PKS* genes without reducing domains [*NR-PKS* genes] are linked to nonreduced compounds such as gyrophoric acid, olivetoric acid [20], physodic acid [20], and grayanic acid [19]), (ii) it must be present in all of the *Umbilicaria* genomes, as all of the species have GA as the major secondary metabolite (33), and (iii) it must be closely related to the *PKSs* involved in the synthesis of orcinol depsides, i.e., PKS16 (19, 20), because orsellinic acid units constitute the building blocks of GA.

**Homology between orcinol-depside clusters.** Homologous GA clusters from *Umbilicaria* spp. were visualized using synteny plots as implemented in Easyfig v2.2.3 (60). In addition, we also inferred the synteny between the GA cluster and the other clusters involved in the synthesis of orcinol derivatives—the olivetoric acid (20), grayanic acid (19), and the orsellinic/lecanoric acid cluster (61, 62). The .gbk input files for Easyfig were taken from antiSMASH (48). Easyfig was run with tblastx v2.6.0+ and a minimum identity value of 50 to draw the blast hits. Clusters were matched for orientation so that the *PKS* genes were oriented in the same direction (Fig. 4).

**BGC clustering: BiG-SCAPE and CORASON.** We used BiG-SCAPE and CORASON (63) to identify the gene cluster networks and infer evolutionary relationships among clusters of interest among different *Umbilicaria* spp. BiG-SCAPE utilizes antiSMASH (48) and MIBiG databases (64) for inferring BGC sequence similarity networks, whereas CORASON employs a phylogenomic approach to infer evolutionary relationships between the clusters. BiG-SCAPE v1.0.1 was run in –auto mode to identify BGC families using antiSMASH output files (.gbk) as input. Networks were generated using similarity thresholds of 0.25. The most likely GA cluster from all of the *Umbilicaria* spp. was examined for conservation and variation among different *Umbilicaria* species using the CORASON pipeline. The antiSMASH .gbk files of the corresponding clusters, based on phylogenetic grouping, were used as input. The most-likely GA cluster from *U. deusta* was used as reference to fish out the most closely related clusters from the other *Umbilicaria* spp.

**Data availability.** This whole-genome shotgun project has been deposited at GenBank under the accession number PRJNA820300. The versions described in this paper are the versions JALILQ000000000 to JALILY000000000. The lichen samples of eight *Umbilicaria* spp. and *Dermatocarpon miniatum* have been deposited under BioSample accession numbers SAMN27294873 to SAMN27294881. The mycobiont samples have been deposited under BioSample accession numbers SAMN26992773 to SAMN26992781. The antiSMASH files of *Umbilicaria* spp. and *Dermatocarpon miniatum* and the candidate gyrophoric acid cluster .gbk files are available at figshare (https://doi.org/10.6084/m9.figshare.19625997).

## SUPPLEMENTAL MATERIAL

Supplemental material is available online only.
**SUPPLEMENTAL FILE 1**, XLSX file, 0.01 MB.
**SUPPLEMENTAL FILE 2**, XLSX file, 0.2 MB.

## ACKNOWLEDGMENTS

This research was funded by LOEWE-Centre TBG, funded by the Hessen State Ministry of Higher Education, Research and the Arts (HMWK).

We thank Daniele Armaleo (Duke University) for his input on the PKS domain composition of tridepsides and didepsides and Marnix Medema and Satria Kautsar for their support with the BiG-SCAPE program.

We thank the reviewers for their constructive suggestions on the manuscript.

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
