## [Reviewer comments · Microbiology Spectrum]

Microbiology Spectrum

A candidate gene cluster for the bioactive natural product gyrophoric acid in lichen-forming fungi

Garima Singh, Anjuli Calchera, Dominik Merges, Henrique Valim, Jürgen Otte, Imke Schmitt, and Francesco Dal Grande

Corresponding Author(s): Garima Singh, Senckenberg Gesellschaft für Naturforschung

Review Timeline:

Submission Date:	January 20, 2022
Editorial Decision:	March 15, 2022
Revision Received:	April 21, 2022
Editorial Decision:	May 16, 2022
Revision Received:	May 19, 2022
Accepted:	June 12, 2022

Editor: Lea Atanasova

Reviewer(s): Disclosure of reviewer identity is with reference to reviewer comments included in decision letter(s). The following individuals involved in review of your submission have agreed to reveal their identity: Ekaterina Shelest (Reviewer #1); Young-Mo Kim (Reviewer #2)

Transaction Report:

DOI: <https://doi.org/10.1128/spectrum.00109-22>

March 15, 2022

Dr. Garima Singh
Senckenberg Gesellschaft für Naturforschung
Frankfurt am Main 60325
Germany

Re: Spectrum00109-22 (A candidate gene cluster for the bioactive natural product gyrophoric acid in lichen-forming fungi)

Dear Dr. Garima Singh:

Thank you for submitting your manuscript to Microbiology Spectrum.

The manuscript underwent a thorough review process and the reviewers agree that the paper lacks substantial information about the data and its analysis, and includes speculative conclusions thus the manuscript in this current form is not publishable in Microbiology Spectrum Journal. However, one of the reviewers would like to see the data and the analysis done as well as substantial improvements on the manuscript thus I would like to give you the opportunity to drastically improve the manuscript showcasing the full data set, include the data that support your conclusions and run additional analysis. Please note that the improved version of the manuscript will undergo another thorough revision process and if the manuscript does not meet the criteria of the journal it might still get rejected without further resubmission. Thus, I encourage you to revise your work beyond the comments of the reviewers.

Link Not Available

Sincerely,

Lea Atanasova
Editor, Microbiology Spectrum

Journals Department
Reviewer comments:

Reviewer #1 (Public repository details (Required)):

There is no information about the submission to a public repository. The data is not accessible for reviewers. I insist on making the data publicly available before the publication.

Reviewer #1 (Comments for the Author):

The paper describes de novo sequencing of several Umbilicaria species with the aim of characterisation of novel BGC. A PKS most likely responsible for the biosynthesis of GA, the most typical NP compound produced by Umbilicaria, was predicted based on phylogenetic analysis. BGCs were predicted using antiSMASH. The clusters contained 9-15-genes. No functions were predicted for any genes apart from PKS itself and occasional P450.

The major problem with this manuscript is almost total lack of data. Genomes are not provided, neither are cluster sequences. I assume that some analysis has been done but not shown (e.g., the structure of the GA PKS). The manuscript is potentially interesting but it's hard to judge about that.

For a proper review, I should have an opportunity to look at the genomes, find the cluster genes, etc. I should have the information about what gene is PKS16, and have the protein sequences of the cluster genes so that I can, if I decide so, look into them. Reviews are not only for criticism, they are also for suggestions. I cannot do my job properly if I lack the data.

So far, here are my remarks:

Major concerns:

antiSMASH BGC predictions are based on similarity search using a database of known BGC genes. That means, that in the majority of cases, the genes in BGCs have known functions. I find it unlikely that in clusters of 9-15 genes no genes could be characterized. As no genomic data is provided, I cannot check the antiSMASH predictions to see if this is really the case or check InterProScan domain predictions. I assume that the authors relied on Blast in their annotation pipeline, which is not always the best way to annotate genes.

In the absence of any relevant information about cluster genes, the cluster discovery is limited just to the number of genes, which is not very impressive. Even that number is not informative as of 9-15 genes in clusters, only 3 are conserved in all considered species, one of them PKS.

L134-136: "The cluster ... associated with GA is the cluster containing PKS16, as 1) it is present in all Umbilicaria spp., 2) it contains an NR-PKS and 3) it forms a monophyletic group with the clade "Group I, PKS 16" from Kim et al (11). - Is it the cluster forming the monophyletic group? Or just the PKSs? If the cluster, where can we see that?"

L.137: The other genes code for unidentified proteins. - Were there indeed no domains in any of the genes? No InterProScan predictions at all? Or did you rely only on the Blast/GO predictions?

L. 145: The BGCs forming a supported monophyletic clade to PKS16 (Group I) - The results of BGC clustering are not shown at all. Should we just trust your word that this is a monophyletic group?

L. 156-158 The BGC associated with the biosynthesis of the following lichen depsides and depsidones have been identified so far: atranorin, lecanoric-, grayanic-, <...>. - haven't you tried to compare your new cluster with these known ones? Especially with the one for lecanoric acid, because you later speculate that your new cluster could be responsible for its biosynthesis, too. I'd assume the PKS described for lecanoric acid should be the same PKS16. Moreover, lecanoric and orcellinic acids are produced not only by lichens (e.g., they've been described in Aspergilli - and their clusters, too). Could be interesting to compare with those PKSs as well.

L. 165: As GA does not have side chain modifications (Fig. 1) we propose that the PKS alone is involved in GA synthesis. - So you propose that the PKS alone is sufficient. Could you speculate what could be functions of other cluster genes? If they were transporters, regulators and other auxiliary genes we usually expect in clusters, they'd be easily recognized as they should have standard domains. So if additional enzymes are not needed and standard genes are not there, what could be the functions of the other 8-12 genes?

L169-173: Our study suggests that the PKSs coding for a didepside and a tridepside differ only in the length of the sequence of the SAT domain. A tridepside PKS contains longer SAT coding sequence than the didepside PKS. The number of ACP and PT domains is the same between the two. - You discuss some results that are not shown at all. This is potentially an interesting part of the study, however, all we can find are PKS sequences in some supplementary table. Even to that there's no reference in the text, I discovered those sequences almost by chance. There is no analysis of the sequences. Are the readers expected to analyse the PKS domain structures on their own?

180-183: Differences among the clusters synthesizing the same compound have been reported before, and have been associated with species-specific ... modifications to the depside released by the PKS. - I far as I understand, GA is not further modified. So this assumption for the differences between GA clusters will not work. Otherwise it should be not GA cluster but the cluster for umbilicarinic acid or hiassic acid, etc.

If it were about regulation, you'd probably discover some TFs. Their domains are known.

L199-200: The lower proportion of hiassic acid as compared to GA could be because the cyt P450 may not be co-expressed with the PKS - No, if you assume the P450 is the part of the cluster. By definition, all genes in a cluster are co-expressed. It does not depend on their orientation (bidirectional promoters are beneficial, but not necessary). It might be, however, that P450 is not a part of the cluster but it's impossible to prove this based only on sequence data, one needs expression data for that. In Fig 4, the choice of the colours is suboptimal. It's almost impossible to distinguish the shades of green of gene 6 (cyt P450) and, for example, gene 10. It's important, because if P450 is not present in several clusters (as it seems), the hypothesis about its involvement in the production of hiassic acid can be easily confirmed just by checked whether the species lacking P450 produce this compound (which is hopefully known, I don't suggest new experiments here).

L. 202: our study provides novel insights into GA cluster composition and organization across different species. - Not so much into the composition, given that no genes are known apart from the PKS.

L.224-227: We propose that the same PKS cluster is most likely involved in the synthesis of GA, umbilicarinic- (an additional methyl group, Fig. 1), hiassic- (additional hydroxyl group, Fig. 1), and lecanoric acid (didepside with no side chains, Fig. 1) in Umbilicaria. - 1. Is it the same PKS as mentioned in the previous sentences? If yes, why there is no comparison with known PKSs? 2. Basically, in the absence of any characterized genes in the cluster your assumptions about its function are based only on the PKS. This is fine but not enough for a paper devoted to a cluster discovery. Given that the major final product is GA (as far as I understood, this is what the strains produce mainly), and it does not need any auxiliary enzymes for its production, it's hard to judge if any cluster is needed at all. As I've mentioned above, MFS, transport proteins, TFs, etc., are all too well known to be missed. If they are not there, I wonder what other cluster functions could be necessary for a product, which only needs a PKS?

L.342: The data used in this study is deposited at XXX - For your information, data must be available for reviewers by the time of review.

Were genomes submitted to any public database? As far as I understand, data availability is one of the prerequisites of a publication.

Minor comments:

L.146/Fig4: The CORASON analysis showed that only three genes on the cluster were shared - Could you mark them in the Fig. 4?

L. 167: The depside PKSs identified so far code for didepsides, i.e. they contain two phenolic rings joined with an ester bond, - sounds like PKSs contain two phenolic rings. Better: i.e., compounds that contain...

L.176: GA cluster contains about 11-15 genes: 11-15 is a concrete range, it's not "about". In addition, the shortest clusters are 9 genes, according to Fig.4.

Fig 4: are white genes non-cluster genes? Should be explained in the legend

For strains of the same species, I'd suggest to remove identical clusters. They do not give any additional information. However, it's up to you.

Reviewer #2 (Comments for the Author):

Singh et al. performed a gene cluster analysis for gyrophoric acid biosynthesis among lichen-forming fungi, and found a few "most-likely" genes. These information will benefit lichen and natural product research communities at some degree, however, the reviewer found several speculations on data interpretation, which needs more evidence to prove (e.g. by actual gene expression, metabolite measurement, activity test and etc.). In addition, grouping information was obtained from the reference paper (Kim et al. 2021), the reviewer doesn't think it is necessary to show the figure 2 as a main figure. Some errors found on citations, so the authors may want to check references thoroughly to make sure they are correctly linked.

Staff Comments:

Preparing Revision Guidelines

Please return the manuscript within 60 days; if you cannot complete the modification within this time period, please contact me. If you do not wish to modify the manuscript and prefer to submit it to another journal, please notify me of your decision immediately so that the manuscript may be formally withdrawn from consideration by Microbiology Spectrum.

Dear Dr. Atanasova,

Thank you very much for sending the reviews of our manuscript.

We appreciate the thorough and constructive suggestions by the reviewers. We have addressed all their comments and provided point-by-point responses to all points raised by them. The line numbers refer to the numbers in the “manuscript-revised-changes-accepted” file.

In particular, the major criticism by Referee #1 concerned i) data availability and ii) lack of information on the function of genes in the gyrophoric acid cluster. As per the Referee’s suggestion, we have submitted the genomes of each *Umbilicaria* species and the *Dermatocarpon miniatum* to NCBI.

In addition, the following data is now submitted to public repositories

- the gyrophoric acid cluster .gbk files (doi: 10.6084/m9.figshare.19625997)
- the results of antiSMASH for the *Umbilicaria* spp. used in the study (doi: 10.6084/m9.figshare.19625997).

The data will be released once the manuscript is published. Meanwhile, we provide the link to access the genomes and other related files for review purpose-only to the reviewer

(<https://drive.google.com/drive/folders/1AzCi5807A-wUFki76gb7EbEXHXNMUfbZ?usp=sharing>)

The amino acid sequences of the gyrophoric acid PKSs used for the phylogenetic analysis are already provided as Supplementary Material.

To address the second point of criticism, we confirmed the functions of the genes using InterProScan as suggested by the Referee. The InterProScan was already performed on the genes but we carefully checked again the log files and the results files to ensure if we missed out some gene annotations.

We have now analyzed the data beyond the suggestion of the reviewers and

- performed CDS search via NCBI CDS search to identify the conserved domains and infer the putative function of all the genes in the putative gyrophoric acid cluster. This analysis is now added to material and methods, results and discussion. The results have been added to Figure 3 (please see 3B).
- added a synteny plot of gyrophoric acid clusters between *Umbilicaria deusta* and other *Umbilicaria* spp. and between the *U. deusta* gyrophoric acid cluster and the other known orcinol-producing clusters identified so far from the lichenized fungi (olivetoric avcid cluster from *Pseudevernia furfuracea* and grayanic acid cluster from *Cladonia grayi*) and non-lichenized fungi (the orsellinic/lecanoric acid cluster of *Aspergillus nidulans*). This plot shows the homology between the PKS genes and also the synteny of the clusters (figure 4).

We believe that the manuscript has greatly benefitted from the Reviewers’ constructive suggestions and we hope that the revised version is now acceptable for publication in *Microbiology Spectrum*.

Sincerely,
Garima Singh

Reviewer #1

- 1. (Public repository details (Required)): There is no information about the submission to a public repository. The data is not accessible for reviewers. I insist on making the data publicly available before the publication.**

We have now submitted the reference genomes of each *Umbilicaria* spp. to NCBI and have already indicated the genome accession numbers in the Data availability section. These will be released at the time of acceptance of the publication. In the meantime, we provide the genomes to the Reviewer for review purpose (<https://drive.google.com/drive/folders/1AzCi5807A-wUFki76gb7EbEXHXNMUfbZ?usp=sharing>).

We provided the amino acid sequences of all PKSs used in the phylogenetic analysis as Supplementary Material 2. Now we also provide the .gbk files of all putative gyrophoric acid clusters. We believe that we now provide access to all the data required to repeat the analyses done in this study for the following reasons:

- We provide the gbk files of the candidate gyrophoric acid clusters for the review purpose. This file contains all the information related to the gyrophoric acid clusters. The data is already submitted to figshare and will be released once the article is published (doi: 10.6084/m9.figshare.19625997).
- We also provide all antiSMASH results. This data is also submitted to figshare and will be released once the article is published (doi: 10.6084/m9.figshare.19625997).
- the Corason results can now be replicated using the .gbk files of the putative GA cluster. This data is also accessible via the provided link.

We have expanded the Data availability section to include the latest information available data as follows:

Lines 385-393

Data Availability

This Whole Genome Shotgun project has been deposited at GenBank under the accession PRJNA820300. The versions described in this paper are the versions JALILQ000000000 - JALILY000000000. The lichen samples of eight *Umbilicaria* Spp. and *Dermatocarpon miniatum* have been deposited as Biosamples SAMN27294873 - SAMN27294881. The mycobiont samples have been deposited as Biosamples SAMN26992773 - SAMN26992781. The antiSMASH files of *Umbilicaria* spp. and *Dermatocarpon miniatum* and the candidate gyrophoric acid cluster gbk files are available at figshare (doi: 10.6084/m9.figshare.19625997).

- 2. The paper describes de novo sequencing of several *Umbilicaria* species with the aim of characterisation of novel BGC. A PKS most likely responsible for the biosynthesis of GA, the most typical NP compound produced by *Umbilicaria*, was predicted based on phylogenetic analysis. BGCs were predicted using antiSMASH. The clusters contained 9-15-genes. No functions were predicted for any genes apart from PKS itself and occasional P450.**

We thank the Reviewer for pointing out the lack of information on the function of genes. We have performed additional analysis to infer the function of genes based on conserved domains. We think that information on the function of the genes is a very informative addition to the manuscript.

First, we clarified the sentence to specify that the genes were of unknown function according to antiSMASH. We have now mentioned that both antiSMASH and InterProScan annotated the genes as coding for hypothetical proteins.

In addition, we performed a CDS search on the individual genes of each cluster. Although antiSMASH and InterProScan annotated only PKS and cyt P450 genes, the CDS search found conserved domains in additional ten out of 15 genes in the cluster. The putative functions of these genes are the following: protein trafficking, oxidation, transcriptional regulation, hydrolases, carrier protein, protein-protein binding, protein-protein interaction, and metabolism. In two genes conserved domains were detected but were identified as DUFs (domain of unknown function). For three genes no conserved domains were detected. We have added details on the CDS search in the Materials and Methods and Results sections to indicate the putative function of the genes.

Materials & methods:

Lines 337-339

To complement the functional annotation of genes by InterProScan, egg-NOG-mapper and SignalP, we performed NCBI CDS searches on the genes of the putative gyrophoric acid cluster using the nucleotide sequences of the genes.

Results:

Lines 138-146

Out of 15 genes (in GA cluster of *U. deusta*, Fig. 3A), antiSMASH identified two genes, PKS and cyt P450, whereas other genes were identified as coding for proteins of unknown function by antiSMASH as well as InterProScan. NCBI CDS identified conserved domains for an additional 10 genes and estimated their putative function based on these domains (Fig. 3 B). Specifically, these genes code for enzymes involved in transcription regulation, oxidation, hydrolysis, and protein-protein interaction/trafficking. Conserved domains were not detected in three genes (gene 5, 11, and 13, Fig. 3B), and two genes (gene 7 and 14, Fig. 3B) had conserved domains belonging to the DUF superfamily (domain of unknown function).

Discussion:

Lines 196-208

The most-likely GA cluster contains about 9-15 genes in different *Umbilicaria* spp. (Fig. 4, 5). Interestingly, only three genes are conserved across the species analyzed: the PKS, a gene having a conserved domain of unknown function present upstream, and a hydrolase present downstream of the PKS (Fig. 4, 5). This suggests that these three genes form an integral part of the GA cluster, whereas the other genes are facultative among GA producing *Umbilicaria* spp. These facultative genes code for enzymes involved in transcription regulation, oxidation, hydrolysis, and protein-protein interaction/trafficking, indicating that they play a role in species-specific transcription regulation, transport or modification (Fig. 3). Differences among the clusters synthesizing the same compound have been reported before. For instance, the usnic acid cluster differs in its gene content among taxa and out of 8-15 genes belonging to the cluster, only PKS and cyt P450 are conserved (26). These differences could be responsible for species-specific BGC regulation or modifications to the protein released by the PKS.

- 3. The major problem with this manuscript is almost total lack of data. Genomes are not provided, neither are cluster sequences. I assume that some analysis has been done but not shown (e.g., the structure of the GA PKS). The manuscript is potentially interesting but it's hard to judge about that.**

We submitted all genomes to the NCBI. The antiSMASH files and the gyrophoric acid cluster gbk files are also submitted to figshare and will be available publicly. With this all the data related to the study will be provided.

The structure of GA PKS, showing the PKS domains, is provided as Figure 3A.

Additionally, we now provide synteny plots to compare the synteny among all putative gyrophoric acid gene clusters found in the studied *Umbilicaria* spp. (figure 4)

For a proper review, I should have an opportunity to look at the genomes, find the cluster genes, etc. I should have the information about what gene is PKS16, and have the protein sequences of the cluster genes so that I can, if I decide so, look into them. Reviews are not only for criticism, they are also for suggestions. I cannot do my job properly if I lack the data.

We now provide the Reviewer with access to the genomes, antiSMASH results and the gyrophoric acid cluster gbk files (files only for review purpose). The genome assembly files do not have gene information so the clusters cannot be found or linked to the BGCs in the genome assembly files. Therefore, we also provide the .gbk files for all the putative gyrophoric acid clusters from all the *Umbilicaria* spp. used in the study. This file contains all the information about the genes of the cluster, including their nucleotide, mRNA, and amino acid as well as CDS sequences.

The protein sequences of the PKSs used for the phylogenetic analysis are provided in the Supplementary Material 2 file. This information is now also available in the .gbk files of all the putative gyrophoric acid clusters.

The GA cluster as predicted by antiSMASH, along with the domains, is given as Figure 3.

Major concerns:

4. antiSMASH BGC predictions are based on similarity search using a database of known BGC genes. That means, that in the majority of cases, the genes in BGCs have known functions. I find it unlikely that in clusters of 9-15 genes no genes could be characterized. As no genomic data is provided, I cannot check the antiSMASH predictions to see if this is really the case or check InterProScan domain predictions. I assume that the authors relied on Blast in their annotation pipeline, which is not always the best way to annotate genes. In the absence of any relevant information about cluster genes, the cluster discovery is limited just to the number of genes, which is not very impressive. Even that number is not informative as of 9-15 genes in clusters, only 3 are conserved in all considered species, one of them PKS.

We now provide the .gbk files containing CDS and protein similarity information. In addition, we have performed NCBI CDS searches on individual genes of a gyrophoric cluster to predict their functions. Indeed, several conserved domains in the genes were identified in the CDS search even though InterProScan identified their product as hypothetical protein. Apart from the PKS and cyt. P450 which were identified by antiSMASH as well as InterProScan, using the CDS search we identified:

- a transporter protein
- a transcriptional regulator
- a carrier protein
- a hydrolase
- an oxidase
- an Acyl CoA binding protein (ACBP)

- a gene coding for a product causing protein-protein interaction
- a gene coding for phosphorylcholine metabolism

For 3 genes no conserved domains were detected. Two genes contained conserved DUF (domain of unknown function) domains.

These results are provided in Figure 3B.

- 5. L134-136: "The cluster ... associated with GA is the cluster containing PKS16, as 1) it is present in all *Umbilicaria* spp., 2) it contains an NR-PKS and 3) it forms a monophyletic group with the clade "Group I, PKS 16" from Kim et al (11). - Is it the cluster forming the monophyletic group? Or just the PKSs? If the cluster, where can we see that?"**

We rephrased the sentence to clarify that PKS16 forms a monophyletic clade. This clade is shown in Figure 2.

- 6. L.137: The other genes code for unidentified proteins. - Were there indeed no domains in any of the genes? No InterProScan predictions at all? Or did you rely only on the Blast/GO predictions?**

We have now provided the information on the domains and putative functions of the other genes of the cluster based on the NCBI CDS search and have included these results in Figure 3. Please refer to point 4 for details.

- 7. L. 145: The BGCs forming a supported monophyletic clade to PKS16 (Group I) - The results of BGC clustering are not shown at all. Should we just trust your word that this is a monophyletic group?**

We have rephrased the sentence to clarify that we mean the monophyletic clade in the phylogenetic tree (Figure 2) which was inferred from the amino acid sequences of the PKS genes present in *Umbilicaria* and several other lichen-forming fungi.

- 8. L. 156-158 The BGC associated with the biosynthesis of the following lichen depsides and depsidones have been identified so far: atranorin, lecanoric-, grayanic-, <...>. - haven't you tried to compare your new cluster with these known ones? Especially with the one for lecanoric acid, because you later speculate that your new cluster could be responsible for its biosynthesis, too.**

We thank the Reviewer for this suggestion. We now provide the synteny plot (Figure 4) to designate the synteny between the gyrophoric acid cluster and the two lichen depside/depsidone clusters that have been identified so far, i.e. olivetoric acid cluster, and grayanic acid cluster and the *Aspergillus nidulans* orsellinic/lecanoric acid cluster.

We have not analyzed the synteny between the atranorin cluster and the gyrophoric acid cluster as atranorin is a β -orcinol derivative, so the corresponding PKS contains a methyltransferase domain and is only distantly related to the PKS involved in the synthesis of orcinol-derived compounds (lecanoric acid cluster, physodic acid cluster, and grayanic acid) which lack a methyltransferase domain in the PKS.

Also, the mechanism of synthesis and the chemistry of the two compounds is very different. The production of atranorin involves three enzymes: a PKS catalyzing the synthesis of the depside, which is then oxidized by cyt P450 and O-methylated by an O-methyltransferase to produce atranorin. Gyrophoric acid production on the other hand does not involve an oxidation/O-methylation step. The structure of gyrophoric acid suggests that the PKS alone is required for its production.

Materials & methods

Lines 350-356

Homologous GA clusters from *Umbilicaria* spp. were visualized using synteny plots as implemented in Easyfig v2.2.3 (55). In addition, we also inferred the synteny between the GA cluster and the other clusters involved in the synthesis of orcinol derivatives- the olivetoric acid (16), grayanic acid (15) and the orsellinic/lecanoric acid cluster (56, 57). The .gbk input files for Easyfig were taken from antiSMASH (58). Easyfig was run with tblastx v2.6.0+, a minimum identity value of 50 to draw the blast hits. Clusters were matched for orientation so that the PKS were oriented in the same direction (Fig. 4).

Results

Lines 147-157

We inferred the synteny of the *U. deusta* GA cluster with the GA clusters of all other *Umbilicaria* spp. to estimate homology between them (Fig. 4). In addition, we also examined the synteny of *U. deusta* GA cluster with the other clusters involved in the synthesis of orsellinic-acid-derivatives compounds– olivetoric acid, grayanic acid and orsellinic acid cluster (Figure 4). The synteny plots show that GA clusters are highly homologous among *Umbilicaria deusta*, *U. freyi*, *U. grisea*, *U. phaea* and *U. subpolyphylla* whereas between GA and grayanic acid cluster and GA and olivetoric acid cluster only PKSs are homologous. The other genes of the clusters involved in the production of orsellinic acid-derivatives not conserved among the genera examined, i.e., *Umbilicaria* spp., *Cladonia grayi*, *Pseudevernia furfuracea*. The orsellinic acid cluster from *Aspergillus nidulans* showed almost no homology to the GA cluster.

Discussion

Lines 228-235

Although the GA cluster shows varied degree of homology among *Umbilicaria* spp., with certain cluster genes present only in some species, gyrophoric acid PKSs are highly homologous among *Umbilicaria* (Fig. 4). This is expected as the PKSs are involved in the synthesis of the same compound. Interestingly, the GA PKS is also homologous to the olivetoric acid PKS and grayanic acid PKS suggesting that the depside/depsidone PKS are conserved even among distantly related taxa in lichen-forming fungi. The GA was not homologous to the orsellinic acid PKS from *Aspergillus nidulans* (Fig. 4) suggesting different evolutionary paths despite being involved in the synthesis of an orcinol-derived compound.

- 9. I'd assume the PKS described for lecanoric acid should be the same PKS16. Moreover, lecanoric and orsellinic acids are produced not only by lichens (e.g., they've been described in Aspergilli - and their clusters, too). Could be interesting to compare with those PKSs as well.**

We have now assessed the synteny between the clusters of other lichen orsellinic acid-derived compounds (grayanic acid and physodic acid clusters) and gyrophoric acid cluster. In addition, we have evaluated the synteny between the gyrophoric acid cluster from different *Umbilicaria* spp. The resulting figure is provided as Figure 4

For details see the point 8.

- 10. L. 165: As GA does not have side chain modifications (Fig. 1) we propose that the PKS alone is involved in GA synthesis. - So you propose that the PKS alone is sufficient. Could you speculate what could be functions of other cluster genes? If they were transporters, regulators and other auxiliary genes we usually expect in clusters, they'd be easily recognized as they should have standard domains. So if additional enzymes are not needed and standard genes are not there, what could be the functions of the other 8-12 genes?**

Yes, we suggest that PKS alone is sufficient for the synthesis of the depside. Indeed, the other genes have a regulatory or transport-related, or interaction-related function as shown by the CDS analysis.

The function of other genes in the candidate gyrophoric acid cluster as estimated by InterProScan and CDS search is now mentioned in the Materials and Methods and Results sections. Most of the other genes in the cluster have a binding, regulatory or transport function. We detected a cyt P450, an oxidase, and a hydrolase. We propose that these genes are involved in the side-chain modifications leading to the production of hiassic acid, and umbilicic acid, other minor metabolites (0.1 to 0.7% dw) detected from *Umbilicaria* species as compared to gyrophoric acid (4-6% dw). Hiassic acid involves an oxidation of gyrophoric acid and umbilicic acid involves an O-methylation.

These results have been added to the manuscript (please see point 4 for details).

It has been shown recently (2021) that PKS alone is sufficient to synthesize the depside. The heterologous expression of lecanoric acid PKS showed that PKS alone was capable of producing the depside (Kealey et al. 2021, Met. Eng. Comm). Lecanoric acid does not have an acyl side chain or other branch modification just like gyrophoric acid. On the other hand, for atranorin (compound with an oxidized and a methylated carbon), PKS synthesizes the depside intermediate/precursor compound 4-O-demethylbarbatic acid and only after the addition of subsequent trailing enzymes such as cyt P450 and O-methyltransferase the final product atranorin was obtained (Kim et al. 2021, mBio).

- 11. L169-173: Our study suggests that the PKSs coding for a didepside and a tridepside differ only in the length of the sequence of the SAT domain. A tridepside PKS contains longer SAT coding sequence than the didepside PKS. The number of ACP and PT domains is the same between the two. - You discuss some results that are not shown at all. This is potentially an interesting part of the study, however, all we can find are PKS sequences in some supplementary table. Even to that there's no reference in the text, I discovered those sequences almost by chance. There is no analysis of the sequences. Are the readers expected to analyse the PKS domain structures on their own?**

We thank reviewer for this suggestion. Indeed, this is very interesting part of the study. We have now added a figure displaying the homology between the tridepside gyrophoric acid PKS and the didepside olivetoric acid and the didepsidone grayanic acid PKSs (Figure 4). These results are now mentioned in the manuscript text as well.

Discussion

Lines 189-193

The depside PKSs identified so far code for didepsides, i.e. compounds that contain two phenolic rings joined with an ester bond, for example, atranorin, and olivetoric acid (13, 15, 16). Ours is the first study to identify the most-likely PKS associated with a tridepside synthesis, i.e., three

phenolic rings joined with two ester bonds. Our study suggests that the PKSs coding for a didepside and a tridepside are highly homologous (Fig. 4).

12. **180-183: Differences among the clusters synthesizing the same compound have been reported before, and have been associated with species-specific ... modifications to the depside released by the PKS. - I far as I understand, GA is not further modified. So this assumption for the differences between GA clusters will not work. Otherwise it should be not GA cluster but the cluster for umbilicic acid or hiassic acid, etc.**

Yes, the GA is not further modified. For instance, it does not have an oxidized side chain or a methylated carbon, two very common types of modifications. The PKS releases the tridepside. As there is only one depside cluster and the *Umbilicaria* spp. produce other depsides as well (hiassic- and umbilicic acid), this is the only candidate cluster potentially producing the major metabolite gyrophoric acid as well as the minor metabolites in *Umbilicaria* spp.

Umbilicic- hiassic acids are only secreted by some species whereas gyrophoric acid is either the major or the only metabolite present in all investigated species that is why we call it the 'gyrophoric acid cluster'.

Therefore, PKS is the gyrophoric acid PKS whereas the cluster is potentially involved in the synthesis of gyrophoric- hiassic-, as well as umbilicic acid depending upon which other genes are expressed; as mentioned above, umbilicic acid would require an O-methyltransferase and hiassic acid would require a cyt P450.

We have clarified this in the manuscript.

This is now also mentioned in the Abstract:

Lines 34-37

In addition, our results suggest that the same cluster codes for different but structurally similar NPs, i.e., GA, umbilicic acid and hiassic acid, bringing new evidence that lichen metabolite diversity is also generated through regulatory mechanisms at the molecular level.

13. **If it were about regulation, you'd probably discover some TFs. Their domains are known.**

Thank you for this comment. We have identified TFs among the genes of the cluster. The gene is now marked and mentioned in Figure 3.

14. **L199-200: The lower proportion of hiassic acid as compared to GA could be because the cyt P450 may not be co-expressed with the PKS - No, if you assume the P450 is the part of the cluster. By definition, all genes in a cluster are co-expressed. It does not depend on their orientation (bidirectional promoters are beneficial, but not necessary). It might be, however, that P450 is not a part of the cluster but it's impossible to prove this based only on sequence data, one needs expression data for that.**

It is known, and rather common - that all the genes of a cluster are not always co-expressed. Here are a few examples:

It has been shown in fungi that only a certain set of genes in a biosynthetic gene cluster are active at a time and that a cluster may produce different compounds depending upon the combination of genes expressed. For instance, the aspyridone cluster in *Aspergillus nidulans*

produces up to eight different compounds depending on the combination of genes activated in the cluster (Wasil et al., Chem Sci 2013).

In another study on chemotypes of *Pseudevernia furfuracea* genomics and expression data showed that out of 10 genes in the olivetoric/physodic acid cluster, only three genes were highly transcribed in the cluster while the other genes were almost transcriptionally silent (Singh et al. 2021, Biomolecules).

Furthermore, it has been shown that some of the genes of a cluster may be regulated by the TFs present outside the cluster (the so-called 'global regulators'). This indicates that all the genes of a cluster are not co-expressed even when the local TFs is co-expressed with the other genes of a cluster. For instance, the deletion of the TF present in the red pigment bikaverin cluster in *Fusarium fujikuroi* leads to loss of expression of only some genes but not all (Wiemann et al. 2009, Mol Microbiol). Thus, the other genes are regulated by an external TF and may not be co-expressed with the other genes of that cluster.

We have modified the text to add more examples (apart from the grayanic acid one) of the compounds which require cyt P450 for the production of the final compound – usnic acid and atranorin. In these clusters as well, the cyt P450 lies next to the PKS and in opposite orientation.

Lines 209-227

The most-likely GA cluster contains a cyt P450 (Fig. 3A), which has been associated with depsidone production via oxidation of an acyl chain (13, 15, 16, 27). However, the location and orientation of the cyt P450 in the putative Umbilicaria spp. GA clusters is different from a cluster which requires an active cyt P450 for the production of the final compound, i.e., grayanic acid, usnic acid and atranorin cluster (Fig. 4) (13, 15, 26). In the GA cluster, the cyt P450 is not located next to the PKS and has the same orientation as PKS, whereas in the grayanic acid, usnic acid and atranorin cluster, cyt P450 lies next to the PKS, in the opposite orientation (Fig. 4). For instance, for grayanic acid synthesis (in *Cladonia grayi*) involves the synthesis of the depside intermediate by PKS followed by oxidation of the released depside into depsidone by cyt P450 (15). Such organization is suggestive of genes being regulated and co-expressed by the same promoter (13, 15). The PKS and cyt P450 form the integral part of depsidone synthesis (34) whereas the depside is coded by the PKS alone, with the exception of the side chain modifications (13, 14). Therefore, despite being part of the GA cluster, the cyt P450 does not seem to be involved in GA synthesis or in the synthesis of umbilicatic- and/or lecanoric acid reported from Umbilicaria spp. analyzed in this study. The synthesis of hiascic acid however would require the hydroxylation of a methyl group by cyt P450 enzyme after the depside is released from the PKS (Fig. 1, the OH group in bold in hiascic acid). The lower proportion of hiascic acid as compared to GA could be because the cyt P450 may not be co-expressed with the PKS.

15. **In Fig 4, the choice of the colours is suboptimal. It's almost impossible to distinguish the shades of green of gene 6 (cyt P450) and, for example, gene 10. It's important, because if P450 is not present in several clusters (as it seems), the hypothesis about its involvement in the production of hiascic acid can be easily confirmed just by checked whether the species lacking P450 produce this compound (which is hopefully known, I don't suggest new experiments here).**

We have changed the colors in the figures to make them well distinguishable from each other.

16. **L. 202: our study provides novel insights into GA cluster composition and organization across different species. - Not so much into the composition, given that no genes are known apart from the PKS.**

Following the Reviewer's suggestions, we have now added information on the function of other genes of the cluster. Please see point 4 for details.

17. L.224-227: We propose that the same PKS cluster is most likely involved in the synthesis of GA, umbilicatic- (an additional methyl group, Fig. 1), hiassic- (additional hydroxyl group, Fig. 1), and lecanoric acid (didepside with no side chains, Fig. 1) in Umbilicaria.

- 1. Is it the same PKS as mentioned in the previous sentences? If yes, why there is no comparison with known PKSs?

2. Basically, in the absence of any characterized genes in the cluster your assumptions about its function are based only on the PKS. This is fine but not enough for a paper devoted to a cluster discovery. Given that the major final product is GA (as far as I understood, this is what the strains produce mainly), and it does not need any auxiliary enzymes for its production, it's hard to judge if any cluster is needed at all. As I've mentioned above, MFS, transport proteins, TFs, etc., are all too well known to be missed. If they are not there, I wonder what other cluster functions could be necessary for a product, which only needs a PKS?

We have added synteny plots (figure 4) to represent the homology between the gyrophoric acid cluster and other known clusters from lichen-forming fungi, i.e., olivetoric /physodic acid and grayanic acid cluster as well as the orsellinic acid cluster of *Aspergillus nidulans*.

We thank the Reviewer for driving our attention to the function of additional genes in the cluster. We have now estimated the functions of other genes using CDS search and we report the functions of these genes in the manuscript.

18. L.342: The data used in this study is deposited at XXX - For your information, data must be available for reviewers by the time of review.

All the data related to this study is now made available. We have also added the genome accession numbers to the Data availability section (genomes will be made public once the article is accepted). Additionally, we provide the data to the Reviewer for review purpose.

Lines 385-393

Data availability

This Whole Genome Shotgun project has been deposited at GenBank under the accession PRJNA820300. The versions described in this paper are the versions JALILQ000000000 - JALILY000000000. The lichen samples of eight Umbilicaria Spp. and Dermatocarpon miniatum have been deposited as Biosamples SAMN27294873 - SAMN27294881. The mycobiont samples have been deposited as Biosamples SAMN26992773 - SAMN26992781. The antiSMASH files of Umbilicaria spp. and Dermatocarpon miniatum and the candidate gyrophoric acid cluster gbk files are available at figshare (doi: 10.6084/m9.figshare.19625997)

19. Were genomes submitted to any public database? As far as I understand, data availability is one of the prerequisites of a publication.

Yes, the genomes are now submitted to ncbi and the accession numbers are provided in the manuscript. They are linked to the manuscript and will be released after the its publication. We have provided the genomes for review purpose to the reviewers.

20. Minor comments:

21. **L.146/Fig4: The CORASON analysis showed that only three genes on the cluster were shared - Could you mark them in the Fig. 4?**

Done. We have marked these genes with a rectangle outside the genes.

22. **L. 167: The depside PKSs identified so far code for didepsides, i.e. they contain two phenolic rings joined with an ester bond, - sounds like PKSs contain two phenolic rings. Better: i.e., compounds that contain...**

Done.

23. **L.176: GA cluster contains about 11-15 genes: 11-15 is a concrete range, it's not "about". In addition, the shortest clusters are 9 genes, according to Fig.4.**

Done. We changed the text to 9-15 genes.

Lines 167-168

CORASON plot displays the GA clusters from *Umbilicaria* spp. (Fig. 5). There are 9-15 genes present in the putative GA cluster of the species' studied.

Lines 196-197

The most-likely GA cluster contains about 9-15 genes in different *Umbilicaria* spp. (Fig. 4, 5).

24. **Fig 4: are white genes non-cluster genes? Should be explained in the legend**

The white genes are the cluster genes but have low homology to the genes of the *U. deusta* GA cluster. This is now explained in the legend.

Line no. 419-423. Figure 5 CORASON-based PKS phylogeny to elucidate evolutionary relationships and cluster organization of GA cluster in *Umbilicaria* spp. The bar plot below depicts the percentage of *Umbilicaria* species in which a particular gene is present. The color of the genes is white when they have low homology to the genes of the other gyrophoric acid clusters.

25. **For strains of the same species, I'd suggest to remove identical clusters. They do not give any additional information. However, it's up to you.**

We prefer to keep the duplicates for the following reasons:

- it has been shown that species with broad ecological range (as the *Umbilicaria* spp. used in this study) may have climate-specific gene clusters (Singh et al. 2021, Env. Microbiology). Therefore, we have chosen to use environmental duplicates (when available) to ensure that we do not miss out on these clusters;
- species may comprise chemotypes and chemotypes may have species-specific clusters. For instance, in *Pseudevernia furfuracea* six clusters were present only in one of the chemotypes (Singh et al. 2021, Biomolecules). Including duplicates is therefore essential to consider intraspecies biosynthetic diversity;
- using two genomes (when available) increases the confidence in our conclusions that we indeed have only one cluster per species as the candidate gyrophoric acid cluster;
- the completeness of the genomes used in this study is between 90-99%. Using two genomes ensures that we are not missing out on a cluster simply because it is not assembled and is not present in the final genome assembly.

To summarize, including duplicates better supports our conclusion that we have only one candidate GA cluster per species. For more on this topic, we would like to point to Vincente et al. 2018, Antibiotics.

Reviewer #2 (Comments for the Author):

- 26. Singh et al. performed a gene cluster analysis for gyrophoric acid biosynthesis among lichen-forming fungi, and found a few "most-likely" genes. This information will benefit lichen and natural product research communities at some degree, however, the reviewer found several speculations on data interpretation, which needs more evidence to prove (e.g. by actual gene expression, metabolite measurement, activity test and etc.).**

Thank you very much for giving us the opportunity to discuss and try to clarify this point. We would like to emphasize the robustness of the integrative approach we used, i.e. combining bioinformatic/genome mining with knowledge on the compound structure/synthesis to identify the candidate gene clusters involved in the synthesis of secondary metabolites.

Natural product research has greatly benefitted from the advancement in genomics (identification of the whole biosynthetic landscape of organisms) and bioinformatics (identification of biosynthetic genes and clusters). Using genomic data to identify the biosynthetic genes, several studies have generated comprehensive phylogenies and linked biosynthetic genes to their respective compounds (see, e.g., Abdel Hameed 2015, Fungal Biology, Pizarro et al 2020 BMC Genomics - usnic acid gene cluster; Ogasawara et al. 2015, PLoS ONE - Frankiamicin A cluster; Singh et al. 2021, Biomolecules - olivetoric/physodic gene cluster; Gerasimova et al. 2022, bioRxiv - atranorin cluster).

This approach is especially beneficial for organisms that are difficult to culture and to genetically modify, such as lichens. Heterologous expression in lichens have been successfully carried out only for two lichen-forming fungi. Interestingly, these studies benefitted from the bioinformatically-predicted gene clusters and PKSs to confirm the functions of candidate biosynthetic genes. For instance, Kealey et al. (2021, Met Eng Comm) heterologously expressed the depside gene cluster predicted by Meiser et al. (2017, Sci Rep.) using antiSMASH on the genomic data.

In addition, we would like to highlight that the importance of our study for the biochemistry of lichenized fungi and fungi in general is several-fold:

- our study provides a comprehensive non-reducing PKS phylogeny, adding 13 novel PKS from nine species to the PKSs involved in the synthesis of orcinol compounds. The most recent phylogeny (Kim et al. 2021) contained 15 PKSs in this clade (i.e., 186% increase).
- a candidate PKS for a tridepside has been identified for the first time and forms a supported clade to the didepside PKSs. This is an important step forward in the biochemical studies of fungi and towards understanding the mechanism of PKS-derived natural product synthesis;
- we would like to emphasize that we used information about the chemical structure of the gyrophoric acid as the background to explain the tridepside synthesis mechanism. On top of this we have added robust phylogenetic evidence to narrow down the candidate gyrophoric acid PKS. Combining these two different approaches, we bring strong evidence that our candidate cluster is involved in gyrophoric acid synthesis.

- 27. In addition, grouping information was obtained from the reference paper (Kim et al. 2021), the reviewer doesn't think it is necessary to show the figure 2 as a main figure.**

Figure 2 is the main figure of the manuscript to show the phylogenetic grouping of the Umbilicaria PKSs. This is the main result of the paper. Here are the major reasons to have this figure in the main manuscript:

-It shows that only one PKS per sample groups in the PKS16 clade indicating there is only one possible gene involved in the synthesis of gyrophoric acid.

- We have expanded the phylogeny from Kim et al. 2021, by adding 13 novel PKS from nine species to the PKSs clade involved in the synthesis of orcinol compounds and 127 PKS in total to the Kim et al. 2021 dataset. Kim et al. 2021's phylogeny included seven lichenized fungal species and we have added additional 10.
- . The most recent phylogeny (Kim et al. 2021) contained 15 PKSs in this clade (i.e., 186% increase).
- This is the first non-reducing PKS phylogeny which includes the sequences of a tridepside PKS and we show that the tridepside PKSs form a supported monophyletic clade to the didepside PKSs.

This figure and the dataset (provided as supplementary material S2) will be useful to expand further the PKS phylogeny of fungi as the new genomes will be sequenced to improve the understanding to PKS evolution.

28. Some errors found on citations, so the authors may want to check references thoroughly to make sure they are correctly linked.

We cross-checked the citations to ensure that they are correctly linked.

May 16, 2022

Dr. Garima Singh
Senckenberg Gesellschaft für Naturforschung
Frankfurt am Main 60325
Germany

Re: Spectrum00109-22R1 (A candidate gene cluster for the bioactive natural product gyrophoric acid in lichen-forming fungi)

Dear Dr. Garima Singh:

We are please to inform you that the reviewer is satisfied with mist of your corrections and improvements. However, there are still some remaining remarks, that the authors shell correct that in the next round of revision.

Link Not Available

Sincerely,

Lea Atanasova

Journals Department
Reviewer comments:

Reviewer #1 (Comments for the Author):

First of all, I must say that I am very pleased to see the improvements made by the authors. Now the manuscript looks as it should, with all data available and more analysis shown and discussed. I appreciate the effort.

I have, however, some further remarks:

Results.

The section about Phylogenetic analysis, at least the beginning of it, needs some rearrangement. You first mention the phylogenies of PKSs and only then tell us that you actually found those PKSs. I would start with the phrase "We identified 110 PKSs...", describe them and then switch to phylogenies. In addition, I'd be curious to see how many PKSs (including reducing ones, if there were any) have been discovered in the genomes (a table in supplementary would suffice). By the way, when you

write that there are 12 on average per genome, this cannot be true given that the other figures are correct. 12x15 would be 180, and 110 PKSs in 15 species give 7 on average.

L.126: Four NR-PKSs were common to all species: PKS15, PKS16, PKS20 and a novel PKS clade. - first, it is not clear how the PKSs were numbered. I feel there is a step missing from both results and methods, and this is the annotation of PKSs. It is not described at all. Those numbers are coming from some other source, because the "novel PKS clade" is marked as "unknown". These small things can be really annoying for a reader. So some short explanations would be beneficial.

Secondly, you list 3 PKSs and a clade ("PKS15, PKS16, PKS20 and a novel PKS clade"). This is stylistically incorrect.

L.128-131: Only one NR-PKS per species formed a supported monophyletic clade with

PKS16 (Group I, i.e., orsellinic acid, depside and depsidone NR-PKSs) (Fig. 2). The most likely NR-PKS for the depsidone grayanic acid and the depsides olivetoric and physodic acid

fall within this PKS clade. - Again, the phrases would make more sense in the inverted order. If you first explain why we should be interested in this particular clade, and then tell us about a single PKS clustering with it, it will be easier to comprehend the idea.

In addition, there is a repetition of the same idea of one PKS per species clustering with PKS16 in the next section (Gyrophoric acid cluster). So this part(s) should be revised.

L.141: domains for an additional 10 genes - delete "an"

L.142: Specifically, these genes code for enzymes involved in transcription regulation... - TFs are not enzymes, same as proteins involved in PPI and trafficking. Just write "proteins".

L.168 of the species' studied - delete " ' "

L.168: PKS and a gene upstream and downstream of it -> PKS and genes...

L. 189-193: I would spend some more words on the PKS structure. The figure 4 is very small and shows the whole cluster.

Nothing wrong with the cluster, but you are discussing specifically the PKS, so it would make more sense to show only PKSs in more detail (for instance, showing not only the % of identity, but also the domain structure). In the first version, you mentioned the length of the SAT domain. If this result stands, it could be reflected in the figure, too.

L.216: For instance, for grayanic acid synthesis - delete "for".

Methods.

The section Identification and Annotations of Biosynthetic Gene Clusters does not describe neither identification, nor annotation of the BGCs. Functional annotation by InterProScan, annotation of secreted proteins, etc., were run for the whole genome and not specifically for the clusters. However, the PKS numbering suggests that there was some hidden annotation based on comparison with other clusters (I guess). This probably deserves some description. On the other hand, as I've mentioned above, it would be interesting to read more about the found clusters: how many, what types of PKSs (reducing-non-reducing), etc. You have this information, so it will not be hard to add it.

Staff Comments:

Preparing Revision Guidelines

Please return the manuscript within 60 days; if you cannot complete the modification within this time period, please contact me. If you do not wish to modify the manuscript and prefer to submit it to another journal, please notify me of your decision immediately so that the manuscript may be formally withdrawn from consideration by Microbiology Spectrum.

Dear reviewer,

Thank you for your suggestions on the revised manuscript.

We have now addressed all your suggestions on the revised version of the manuscript. In particular we have added a Table (table 2) listing the details of biosynthetic gene clusters of Umbilicaria and PKSs. In addition, we have also made additional minor changes mostly related to the typos. All the changes are highlighted in the track changes file.

Please find below the point by point response to all the suggestions.

Thank you and best regards,
Garima Singh

comments:

Reviewer #1 (Comments for the Author):

First of all, I must say that I am very pleased to see the improvements made by the authors. Now the manuscript looks as it should, with all data available and more analysis shown and discussed. I appreciate the effort.

I have, however, some further remarks:

Results.

- 1. The section about Phylogenetic analysis, at least the beginning of it, needs some rearrangement. You first mention the phylogenies of PKSs and only then tell us that you actually found those PKSs. I would start with the phrase "We identified 110 PKSs...", describe them and then switch to phylogenies.**

We thank the Reviewer for the suggestion. We have moved the above-mentioned sentence to the beginning of the paragraph. The paragraph has been restructured according to the other suggestions and now reads as follows:

Materials & methods Lines 339-348

Phylogenetic analyses

To search for PKSs involved in the synthesis of GA, we extracted the amino acid sequences of all the NR-PKS from the BGCs predicted by the antiSMASH for the Umbilicaria spp. and Dermatocarpon miniatum (Supplementary Table S2). We then performed a phylogenetic analysis by incorporating these amino acid sequences into the most comprehensive PKS dataset currently available (Supplementary Table S2) (13, 16). This dataset comprises NR-PKS sequences of the following species downloaded from previous publications and public databases: Cladonia borealis, C. grayi, C. macilenta, C. metacorallifera, C. rangiferina, C. uncialis, Pseudevernia furfuracea, Stereocaulon alpinum and Umbilicaria muhlenbergii. The final dataset contains amino acid sequences of 228 NR-PKSs from 18 species belonging to five LFF genera.

Results, Lines 119-128

Total BGCs and Phylogenetic analysis

A total of 406 BGCs and 236 PKSs were identified in 15 Umbilicaria genomes, representing nine species (Table 2). Out of 236, 122 were NR-PKSs, 86 were reducing PKSs, 16 were type-III PKS whereas 12 were fragmented or unclassified because the core PKS was fragmented and the characteristic domains were missing (Table 2).

Four NR-PKSs were common to all species, namely PKS15, PKS16, PKS20 and a novel PKS (forming a monophyletic, supported clade with PKS33; Fig. 2). Only one NR-PKS per species formed a supported monophyletic clade with PKS16 (Group I, i.e., orsellinic acid, depside and depsidone NR-PKSs) (Fig. 2). No PKS from *Dermatocarpon miniatum* grouped within the PKS16 clade, which is expected as *D. miniatum* does not produce orsellinic acid-based compounds.

- 2. In addition, I'd be curious to see how many PKSs (including reducing ones, if there were any) have been discovered in the genomes (a table in supplementary would suffice).**

We have now added a table (Table 2) to include this information.

- 3. By the way, when you write that there are 12 on average per genome, this cannot be true given that the other figures are correct. 12x15 would be 180, and 110 PKSs in 15 species give 7 on average.**

We have clarified this to state that there are nine species and 15 genomes: $110/9=12.23$. There are more genomes than species as we preferred to use duplicates whenever possible to include possible intraspecific variability in BGC content, thus increasing the confidence in the dataset and the results of the analysis.

- 4. L.126: Four NR-PKSs were common to all species: PKS15, PKS16, PKS20 and a novel PKS clade. - first, it is not clear how the PKSs were numbered. I feel there is a step missing from both results and methods, and this is the annotation of PKSs. It is not described at all. Those numbers are coming from some other source, because the "novel PKS clade" is marked as "unknown". These small things can be really annoying for a reader. So some short explanations would be beneficial.**

We thank the reviewer for pointing this out. Agreeing with the reviewer that elaborating on PKS annotation would increase the clarity of the manuscript, we have added the following section to the materials and methods section:

Lines 354-370

Annotation of PKSs

Umbilicaria PKSs were named according to the clustering with pre-annotated PKSs in the phylogeny. NR-PKSs have been categorized into nine groups based on phylogenetic clustering and broad category of the protein coded by them (13). For instance, Group I comprises PKSs involved in the synthesis of zearalenone, orsellinic acid and its derivative compounds and Group II consists of PKSs coding for melanins.

Each of the nine groups contains several PKSs based on the supported phylogenetic clades and protein sequence similarity. For this study, we included

the following 25 NR-PKSs: PKS1, PKS2, PKS8, PKS9, PKS13 to PKS28 (total 16), PKS30 to PKS34 (total 5). Each PKS represents a supported monophyletic clade within a Group in the NR-PKS phylogeny (13). To summarize, Umbilicaria PKSs were annotated and named according to the phylogenetic clustering with the pre-annotated NR-PKS sequences of Cladonia spp. Pseudevernia furfuracea, and Stereocaulon alpinum, downloaded from previous publications and public databases (13, 15, 16). PKS16 and PKS23 have been suggested to be involved in the synthesis of depsides, the chemical category of GA. The PKSs responsible for the synthesis of β -orcinol depsides such as atranorin are PKS23 whereas those involved in the synthesis of orcinol-depsides, such as grayanic acid and olivetoric acid are PKS16 (15, 16). GA is an orcinol depside therefore the corresponding PKS/s would be PKS16.

5. Secondly, you list 3 PKSs and a clade ("PKS15, PKS16, PKS20 and a novel PKS clade"). This is stylistically incorrect.

Done.

The sentence now reads:

Lines 123-124

Four NR-PKSs were common to all species, namely PKS15, PKS16, PKS20 and a novel PKS (forming a monophyletic, supported clade with PKS33; Fig. 2).

6. L.128-131: Only one NR-PKS per species formed a supported monophyletic clade with PKS16 (Group I, i.e., orsellinic acid, depside and depsidone NR-PKSs) (Fig. 2). The most likely NR-PKS for the depsidone grayanic acid and the depsides olivetoric and physodic acid fall within this PKS clade. - Again, the phrases would make more sense in the inverted order. If you first explain why we should be interested in this particular clade, and then tell us about a single PKS clustering with it, it will be easier to comprehend the idea.

We have rephrased the sentence as suggested:

Lines 367-370

The PKSs responsible for the synthesis of β -orcinol depsides such as atranorin are PKS23 whereas those involved in the synthesis of orcinol-depsides, such as grayanic acid and olivetoric acid are PKS16 (15, 16). GA is an orcinol depside therefore the corresponding PKS/s would be PKS16.

7. In addition, there is a repetition of the same idea of one PKS per species clustering with PKS16 in the next section (Gyrophoric acid cluster). So this part(s) should be revised.

We have omitted the repetition.

8. L.141: domains for an additional 10 genes - delete "an"

Done

9. L.142: Specifically, these genes code for enzymes involved in transcription regulation... - TFs are not enzymes, same as proteins involved in PPI and trafficking. Just write "proteins".

Done

10.L.168 of the species' studied - delete " " "

Done

11.L.168: PKS and a gene upstream and downstream of it -> PKS and genes...

Done

12.L. 189-193: I would spend some more words on the PKS structure. The figure 4 is very small and shows the whole cluster. Nothing wrong with the cluster, but you are discussing specifically the PKS, so it would make more sense to show only PKSs in more detail (for instance, showing not only the % of identity, but also the domain structure). In the first version, you mentioned the length of the SAT domain. If this result stands, it could be reflected in the figure, too.

The domain structure of the GA PKS is shown in Figure 3. Since the PKSs of all the depside PKSs are highly homologous, this reflects that the domain structure is also very similar. To make this more obvious we have added a sentence in the legend explaining the domain similarity between the depside PKSs.

Lines 434-439

Figure 4 Synteny plots based on tBLASTn showing the homology and synteny between the putative gyrophoric acid clusters *U. deusta* and other *Umbilicaria* spp. (A) and between the gyrophoric acid cluster from *U. deusta* and grayanic acid cluster from *Cladonia grayi*, olivetoric acid cluster from *Pseudevernia furfuracea* and orsellinic acid cluster from *Aspergillus nidulans* (B). All the PKSs are highly homologous to the GA PKS and have the same domains as the GA PKS: SAT-KS-AT-PT-ACP-ACP-TE.

We now mention the length of different domains in Fig 3.

13.L.216: For instance, for grayanic acid synthesis - delete "for".

Done

14. Methods.

The section Identification and Annotations of Biosynthetic Gene Clusters does not describe neither identification, nor annotation of the BGCs. Functional annotation by InterProScan, annotation of secreted proteins, etc., were run for the whole genome and not specifically for the clusters. However, the PKS numbering suggests that there was some hidden annotation based on comparison with other clusters (I guess). This probably deserves some description.

We have added a section on PKS annotation. Please refer to point4.

15. On the other hand, as I've mentioned above, it would be interesting to read more about the found clusters: how many, what types of PKSs (reducing-non-reducing), etc. You have this information, so it will not be hard to add it.

We agree that this information might be interesting for the readers. We have now added a table (Table 2) to include this information in the manuscript.

We have also included a short text in Results section

Lines 119-122

Total BGCs and Phylogenetic analysis

A total of 406 BGCs and 236 PKSs were identified in 15 Umbilicaria genomes, representing nine species (Table 2). Out of 236, 122 were NR-PKSs, 86 were reducing PKSs, 16 were type-III PKS whereas 12 were partial or unclassified because the core PKS was fragmented and the characteristic domains were missing (Table 2).

June 12, 2022

Dr. Garima Singh
Senckenberg Gesellschaft für Naturforschung
Frankfurt am Main 60325
Germany

Re: Spectrum00109-22R2 (A candidate gene cluster for the bioactive natural product gyrophoric acid in lichen-forming fungi)

Dear Dr. Garima Singh:

I am pleased to inform you that your manuscript has been accepted, and I am forwarding it to the ASM Journals Department for publication. You will be notified when your proofs are ready to be viewed.

Sincerely,

Lea Atanasova
Editor, Microbiology Spectrum

Journals Department
Supplemental Material: Accept
Supplemental Material: Accept